



# Rapid dose rate estimation for trapped charge dating using pXRF measurements of potassium concentration

Sam Woor[1,2], Mitch K. D'Arcy[1], Olav B. Lian[2], Maria Schaarschmidt[2], Julie A. Durcan[3]

[1]Department of Earth, Ocean and Atmospheric Sciences, University of British Columbia, Vancouver, BC V6T 1Z4, Canada
[2]Department of Geoscience, University of the Fraser Valley, Abbottsford, BC V27 7M7, Canada
[3]School of Geography and the Environment, University of Oxford, Oxford, OX1 3QY, UK

*Correspondence to*: Sam Woor (Samuel.Woor@ufv.ca)

**Abstract.** Quantifying environmental radiation dose rates is an essential step in age calculation using trapped charge dating methods. A means of rapid dose rate estimation would therefore be useful for a variety of reasons, especially in contexts where
rapid equivalent dose estimates are available. For instance, for informing sampling strategy, providing initial age estimates, or supporting portable luminescence studies. However, high-precision methods often used for calculating dose rates are typically time consuming and expensive and are impractical for such 'range-finder' applications. Portable X-ray fluorescence (pXRF) offers a rapid means of measuring the Potassium (K) concentration of sediment, although the other radionuclides typically used to calculate dose rates (Uranium (U) and Thorium (Th)) fall beneath its detection limits at the quantities at which they
are usually present in sediments. In this study, we investigate whether pXRF measurements of K concentration alone can be used to accurately estimate total environmental dose rates. A large, global training dataset of 1473 radionuclide samples is used to generate a set of linear relationships between (1) K concentration and external beta dose rate; (2) external beta and gamma dose rates; and (3) external gamma and alpha dose rates. We test the utility of these relationships by measuring the K contents of 67 sediment samples with independent high-precision radionuclide data from a variety of contexts using pXRF.
The resulting K concentrations are then converted to external dose rate estimates using the training equations. A simplified set of attenuation parameters are used to correct infinite matrix dose rate estimates, and these are combined with cosmic ray and internal contributions to rapidly calculate total environmental dose rates for a range of theoretical, common luminescence dating scenarios (such as 180-250 μm quartz that has undergone etching). Results show that pXRF can accurately measure K concentrations in a laboratory setting. The training equations can predict external beta dose rates accurately based on K content
alone, whilst external alpha dose rates are predicted less accurately. In combination, total estimated dose rates show good agreement with their counterparts calculated from high-precision methods, with 68-98% of our results lying within ±20% of unity depending on the scenario. We report better agreement for scenarios where alpha contributions are assumed to be negligible (e.g., in the case of etched, coarse-grained quartz or potassium feldspar). The use of simplified attenuation factors to correct estimated infinite matrix dose rates does not contribute significantly to resulting scatter, with uncertainties mostly
resulting from the training equations. This study serves as a proof of concept that pXRF measurements, along with a set of linear equations and a simplified correction procedure, can be used to rapidly calculate range-finder environmental dose rates.





## 1 Introduction

Trapped charge dating methods such as luminescence and electron-spin resonance dating can be used to determine the time since burial of mineral grains. Age calculation using these methods requires two parameters to be quantified: (1) The equivalent

dose ($D_e$), the amount of radiation dose absorbed by the mineral throughout the burial period, measured in Gray (Gy); and (2) The environmental dose rate ($\dot{D}$), the rate at which environmental radiation is emitted by the surrounding sediment matrix and received from cosmic rays, measured in Gy per time unit, e.g., Gy/a or Gy/ka. Time since burial is thus calculated by:

$$\text{Age} = \frac{D_e}{\dot{D}} \quad \text{(Equation 1)}$$


To determine $\dot{D}$, various individual dose contributions are calculated and summed:

$$\dot{D} = \dot{D}_\alpha + \dot{D}_\beta + \dot{D}_\gamma + \dot{D}_i + \dot{D}_c \quad \text{(Equation 2)}$$

Where $\dot{D}_\alpha$, $\dot{D}_\beta$ and $\dot{D}_\gamma$ are the dose rate contributions of alpha (α) and beta (β) particles and gamma (γ) ray emissions from the sediment matrix external to the mineral grains being dated, respectively; and $\dot{D}_c$ is the contribution from cosmic rays bombarding the Earth. The $\dot{D}_i$ is the sum of contributions from α and β particles arising from decay processes from sources internal to the mineral grains.

The $\dot{D}_\alpha$ results from the decay chains of Th and U, and $\dot{D}_\beta$ and $\dot{D}_\gamma$ from K, Th and U in the surrounding sediment matrix (Guerin et al., 2011). In most luminescence dating studies, internal α contributions are assumed to be either negligible (e.g., Duller, 1992) or an assumed value is provided (e.g., Mejdahl, 1987; Olley et al., 2004). Internal β contributions are usually calculated using assumed concentrations of the internal K contents (e.g., $12.5 \pm 0.5\%$ or $10 \pm 2\%$; Huntley and Baril, 1997; Smedley et al., 2012, respectively) when potassium-rich feldspar (KF) is the mineral being dated. Both the external and internal dose rate

contributions are calculated using the infinite matrix (IM) assumption: that within the surrounding sediment, the rate of energy emitted over the range of interest is equal to the rate of absorption (Guerin et al., 2012). During dose rate calculation, individual IM dose rates are adjusted for a range of attenuating factors, including grain size, water content, and the effectiveness of α particles to ionize mineral crystals (e.g. Durcan et al., 2015 and references therein), all of which influence radiation emission and absorption. The $D_C$ is calculated mathematically from the latitude, longitude, altitude, burial depth and overburden density

of samples, using the equations of Prescott and Hutton (1994).

Typically, the calculation of $D_e$ and $\dot{D}$ require time-consuming and costly laboratory-based sample preparation and measurements. External $\dot{D}_\alpha$, $\dot{D}_\beta$ and $\dot{D}_\gamma$ contributions to $\dot{D}$ are determined using either geochemical measurements of the K, Th and U concentrations within surrounding sediment, or via direct emission counting. Geochemical measurements are carried



out using laboratory methods, such as inductively coupled plasma mass spectrometry (ICP-MS) or neutron activation analysis (NAA, e.g., Woor et al., 2023; Wolfe et al., 2023). Laboratory-based emission counting techniques include thick-source alpha counting (TSAC; e.g., Huntley et al., 1986; Hossain et al., 2002) but emission counting can also be carried out in the field during sample collection using equipment such as portable gamma spectrometers (e.g., Woor et al., 2023). Whilst accurate, these methods typically take hours to weeks, and time or cost restraints can limit sample throughput (e.g., in the case of sending
samples to specialist laboratories for high-precision geochemistry).

The ability to rapidly and inexpensively assess $\dot{D}$ is useful in a variety of contexts. Numerous studies have shown that ages can be estimated by rapidly calculating $D_e$ following truncated sample processing (e.g., skipping the usual mineral separation steps) or by running smaller numbers of sub-sample aliquots than is typical (e.g., Roberts et al., 2009; Durcan et al., 2010).
Such 'range-finder' dating approaches enable the rapid generation of geochronological data, establishing initial age control that can help refine sampling strategy or identify samples of interest for further laboratory preparation (Roberts et al., 2009; Durcan et al., 2010; Leighton and Bailey, 2015; Alexanderson and Bernhardson, 2016). Moreover, over recent years, the use of portable optically stimulated luminescence (pOSL) readers has increased, offering rapid measurements of photon emission in response to optical stimulation in the field (Sanderson and Murphy, 2010). Signals from pOSL readers have been applied in
a variety of geomorphological and archaeological studies (e.g., Bateman et al., 2015; Gray et al., 2018; Stone et al., 2019, 2024; Munyikwa et al., 2021; Rizza et al., 2024) and offer high sample throughput. Environmental dose rates are a key control on pOSL signals (Munyikwa et al., 2021), and therefore being able to rapidly estimate their variability between samples, at least in a relative sense, would be a significant advantage for interpreting pOSL data. Rapid and portable $\dot{D}_{\beta}$ and $\dot{D}_{\gamma}$ determination would also help to assess dose heterogeneity during field sampling, which can arise in complex sedimentary
contexts where IM assumptions do not hold, such as where samples are taken close to stratigraphic boundaries or in heterogeneous rock slices (e.g. Nathan et al., 2003; Smedley et al., 2020; Ou et al., 2022).

Although range-finder dating studies have shown promising results for the rapid determination of De, less attention has been paid to the rapid measurement of $\dot{D}$. Previous work has shown that $\dot{D}$ can be determined in a matter of hours using laboratory-
based emission counting methods (e.g., Ankjægaard and Murray, 2007; Durcan et al., 2010). Ou et al. (2022) also showed that the K concentrations of rock slices used in luminescence dating can be measured accurately with portable X-ray fluorescence (pXRF), and that there is a strong positive correlation between their K contents and their $\dot{D}_{\beta}$ (measured independently using thick source β counting). Portable XRF is designed to measure the elemental concentrations of materials in the field (Lemiere, 2018), so it could have great potential for rapidly and portably estimating $\dot{D}$. However, whilst pXRF can readily determine K
concentrations at magnitudes typical of sediments in luminescence dating studies with an optimized detection limit of 0.005% (Fig.1a; Hall et al., 2014), the normal limits of detection and quantification of U and Th (~3 and 10 ppm, respectively) are typically too high for most sedimentary settings (Fig.1b; Melquiades et al., 2024).





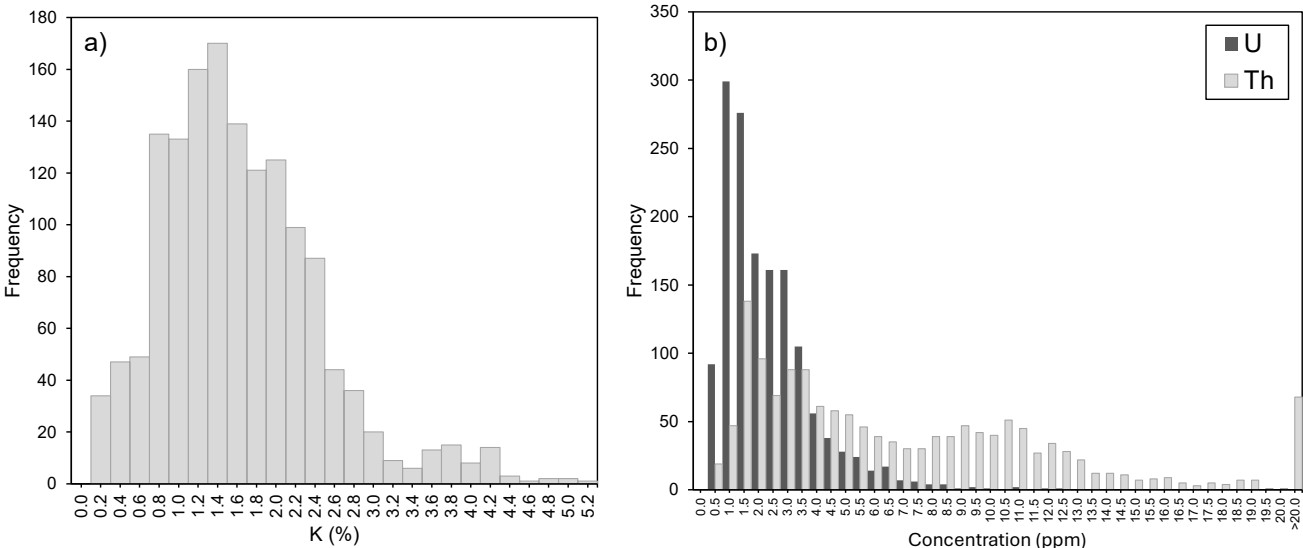

**Figure 1: Histograms of a) K concentrations and b) U and Th concentrations from sediments included in the training dataset compiled for this study (n = 1473; see supplementary information to access the dataset).**

In this study, we develop a method for rapidly estimating range-finder $\dot{D}$ by measuring solely the K concentration of sediments using a laboratory-based pXRF. Like the approach of Ou et al. (2022), this method is based on the relationship between K concentrations and $\dot{D}_\beta$, which is expanded upon to estimate $\dot{D}_\alpha$ and $\dot{D}_\gamma$ using a set of linear equations generated from a large, global sediment radionuclide dataset. These training relationships are used to estimate IM $\dot{D}_\alpha$, $\dot{D}_\beta$ and $\dot{D}_\gamma$ contributions based on K concentrations measured using pXRF for samples with known radionuclide contents. Resulting IM dose rates are given simplified mathematical treatments for attenuation and compared with dose rates calculated based on radionuclide concentrations measured using high-precision geochemistry and corrected using typical attenuation procedures. We demonstrate that it is possible to rapidly estimate $\dot{D}$ with reasonable accuracy and precision by using pXRF-derived K concentrations only, a set of simple linear equations, and a streamlined attenuation approach.

## 2 Methods

### 2.1 Scaling relationships

To estimate $\dot{D}$ based on the K concentration alone, we first establish and test three scaling relationships:

$\dot{D}_\beta \propto K$ (Equation 3)

$\dot{D}_\gamma \propto \dot{D}_\beta$ (Equation 4)

$\dot{D}_\alpha \propto \dot{D}_\gamma$ (Equation 5)

Equation 3 is the positive correlation between IM $\dot{D}_\beta$ and the K concentration of sediment, as has been previously demonstrated (Ankjægaard and Murray, 2007; Roberts et al., 2009; Ou et al., 2022). Equation 4 is the positive correlation between IM $\dot{D}_\gamma$



and IM $\dot{D}_\beta$. Ankjægaard and Murray (2007) showed that IM $\dot{D}_\gamma$ can be estimated from IM $\dot{D}_\beta$ using either a second order polynomial regression or a ratio of ∼0.50 (determined from the slope of a linear fit), from a large suite of luminescence dating

samples (n = 3758). Roberts et al. (2009) produced very similar results using linear regression, with a ratio of 0.59 (n = 427). Lastly, we hypothesize that IM $\dot{D}_\alpha$ scales with IM $\dot{D}_\gamma$ (Equation 5) because α particles are contributed from the U and Th decay chains (not K), and IM $\dot{D}_\gamma$ scales strongly with U and Th concentration (supplementary Fig.S1g, h; Guerin et al., 2011). Therefore, the greater the U and Th concentration, the greater the IM $\dot{D}_\gamma$ and, by extension, the IM $\dot{D}_\alpha$. Using these scaling relationships, we hypothesize that it is possible to estimate IM $\dot{D}_\alpha$, $\dot{D}_\beta$ and $\dot{D}_\gamma$, and therefore $\dot{D}$, from an initial input of the K

concentration.

### 2.1.1 Training dataset

To establish regression relationships and parameterise Equations 3-5, we built a global training dataset of K, U and Th concentrations from published luminescence dating studies, projects undertaken at the University of the Fraser Valley's Luminescence Dating Laboratory and previous compilations of $\dot{D}$ data (Fig.2; Durcan et al., 2015; Woor et al., 2022; Walsh

et al., 2023). The resulting dataset comprises 1473 samples from geographic locations around the world with a broad range of K, U and Th concentrations (Fig.1, Table 1; see supplementary information for the full dataset, including information for calculating $\dot{D}_c$). Infinite-matrix $\dot{D}_\alpha$, $\dot{D}_\beta$ and $\dot{D}_\gamma$ were calculated from these radionuclide data in the Dose Rate and Age Calculator (DRAC; Durcan et al., 2015), using the conversion factors of Guerin et al. (2011). Linear regression was used to parameterise Equations 3-5.

**Table 1: Descriptive statistics of the radionuclide concentrations included within the training dataset. Concentrations are given in % for K and ppm for U and Th (n = 1473). The geographical distributions of samples can be seen in Figure 2 and frequency distributions of radionuclide concentrations in Figure 1.**

| Radionuclide | Mean | Standard deviation | Min. | Max. |
|:---:|:---:|:---:|:---:|:---:|
| K (%) | 1.52 | 0.82 | 0.004 | 5.03 |
| U (ppm) | 2.09 | 1.56 | 0.020 | 12.40 |
| Th (ppm) | 7.17 | 7.18 | 0.030 | 59.00 |




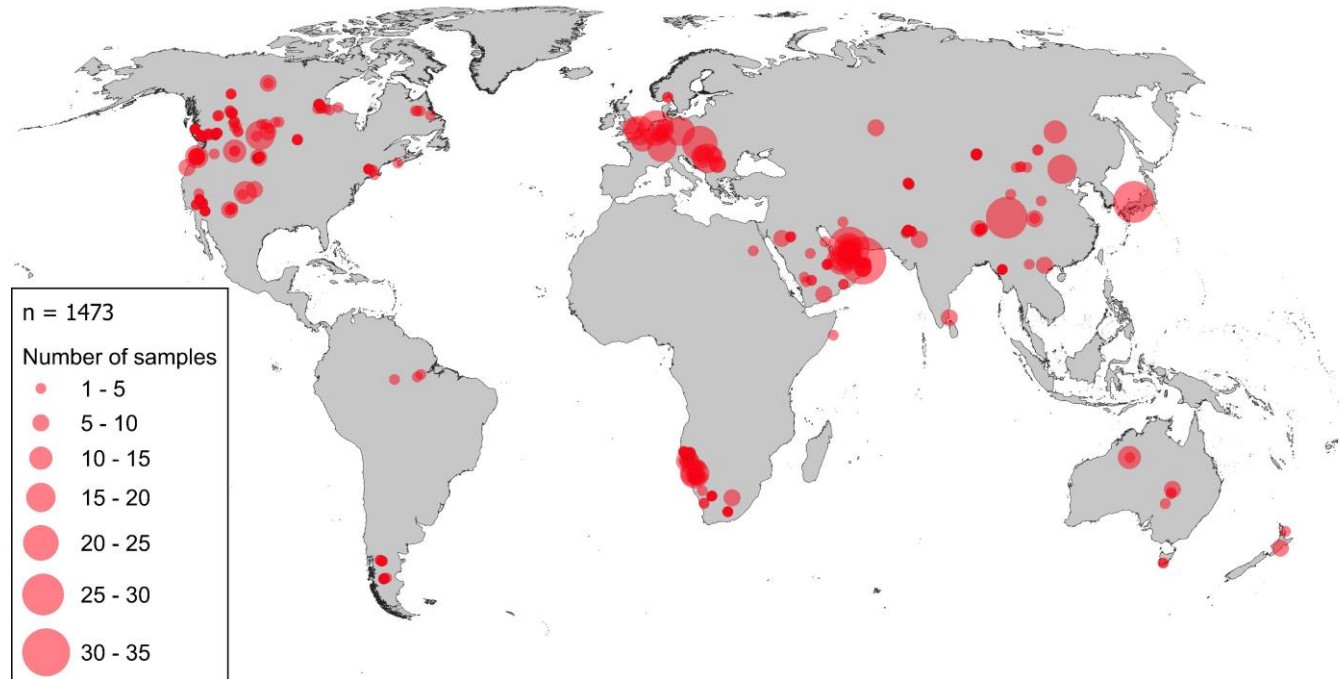

**Figure 2: Map of the sedimentary radionuclide samples compiled within the training dataset used in this study.**

## 2.2 pXRF measurements of K concentrations

Portable XRF was used to measure the K concentrations of 67 sediment dosimetry samples available from the University of the Fraser Valley's Luminescence Dating Laboratory, for which K, U and Th concentrations have previously been measured with NAA or ICP-MS (sample locations and radionuclide concentrations are provided in the supplementary material).

Sediments were oven dried and finely milled prior to packing into cups for analysis. Measurements were carried out using a bench-mounted Olympus Vanta pXRF (Fig.3), with each measurement taking ∼90 s. The pXRF system was operated in two beam mode and each sample was measured three times with the beam hitting different areas of the sediment surface. Throughout the measurements, five certified reference materials with known elemental concentrations and an analytical blank were measured five times each to ensure there was no contamination in the system. The system was cleaned with an air duster

between each measurement.

Resulting pXRF K concentrations were expressed as a percentage and averaged (n = 3) for each of the 67 samples. The measurements were then compared with K concentrations determined using high-precision geochemistry (ICP-MS or NAA) to assess the accuracy of pXRF measurements.




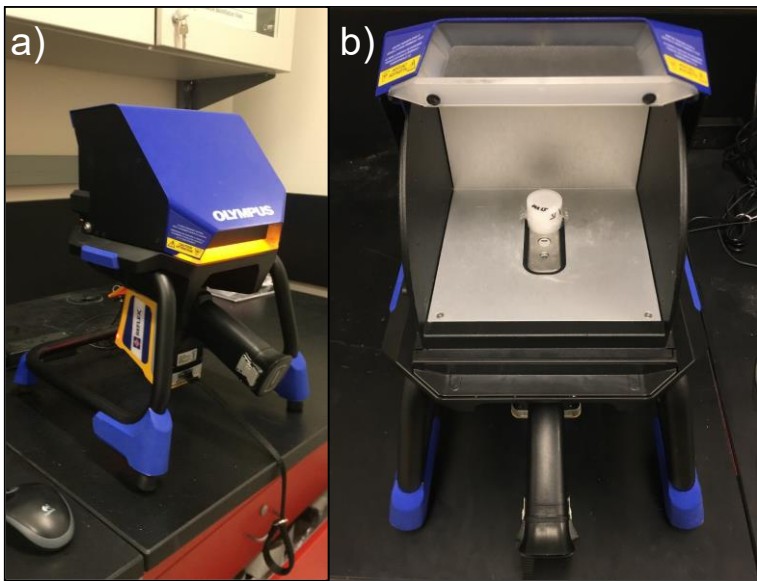

**Figure 3: a) The pXRF in its bench mount with the X-ray shield closed during sample measurement. b) A sample loaded into a cup for analysis placed inside the pXRF's measurement chamber. For scale, the sample is ~2.5 cm in diameter.**

## 2.3 Dose rate calculations

### 2.3.1 High-precision dose rates

To test the accuracy of the rapid, pXRF approach to estimating IM dose rates, total $\dot{D}$ was calculated for the same 67 sediment samples using their high-precision radionuclide contents. Total $\dot{D}$ was calculated for five common, theoretical luminescence dating targets: (1) 180-250 μm quartz (that had undergone etching, the removal of the α-irradiated outer portion of the grain with hydrofluoric acid); (2) 180-250 μm KF (etched); (3) 180-250 μm KF (not etched); (4) 4-11 μm quartz; and (5) 4-11 μm polymineral grains.

The radionuclide conversion factors used to transform radionuclide concentrations into IM dose rates, the attenuation factors used to correct the IM dose rates (grain size, etch depth, grain size, α and β attenuation, α efficiency and water content), the assumptions relating to $\dot{D}_i$ (where applicable), and the parameters used to calculate $\dot{D}_c$ using the equations of Prescott and Hutton (1994) are summarized in Table 2 for each of these theoretical targets. An arbitrary water content of $5 \pm 2\%$ was used to correct dry dose rates using the method of Zimmerman (1971). The contribution of internal α particles was assumed to be negligible in all cases. All dose rate calculations were carried out using DRAC and uncertainties propagated in quadrature (Durcan et al., 2015). All data are available in the supplementary information.





**Table 2: Summary of the parameters and assumptions used to calculate high precision Ḋ for a suite of theoretical luminescence**
**dating targets. Calculations were carried out using DRAC (Durcan et al., 2015).**

| Dose contribution | Input parameter | 180-250 μm quartz | 180-250 μm K-feldspar (etched) | 180-250 μm K-feldspar (not etched) | 4-11 μm polymineral | 4-11 μm quartz |
|---|---|---|---|---|---|---|
| External and internal dose rates | IM $\dot{D}_\alpha$, $\dot{D}_\beta$ and $\dot{D}_\gamma$ (Gy/ka) | Calculated from known radionuclide contents using the conversion factors of Guerin et al. (2011) | | | | |
| | Internal K (%) | NA | 12.5±0.5 (Huntley and Baril, 1997) | | | NA |
| | Min. grain size (μm) | 180 | | | 4 | |
| | Max. grain size (μm) | 250 | | | 11 | |
| | Alpha grain size attenuation | Brennan et al. (1991) | | | | |
| | Beta grain size attenuation | Guerin et al. (2012): values for quartz and feldspar, respectively | | | | |
| | Min. etch depth (μm) | 8 | | NA | | |
| | Max. etch depth (μm) | 10 | | NA | | |
| | Beta etch depth attenuation | Bell (1979) | | | | |
| | α-value | NA | | 0.15±0.05[a] | 0.086±0.004[b] | 0.03±0.003[c] |
| | Water content (%) | 5±2 | | | | |
| Cosmic ray dose rate | Latitude (decimal degrees) | As measured during sampling | | | | |
| | Longitude (decimal degrees) | As measured during sampling | | | | |
| | Altitude (m asl) | As measured during sampling | | | | |
| | Depth (m) | As measured during sampling (±0.05) | | | | |
| | Overburden density (g/cm³) | 1.8±0.1 | | | | |

[a] Value from Balescu and Lamothe (1994).

[b] Value from Rees-Jones (1995).

[c] Value from Mauz et al. (2006).





### 2.3.2 Rapid dose rates

The statistical relationships derived from the training dataset were used to convert pXRF K measurements into IM $\dot{D}_\alpha$, $\dot{D}_\beta$ and $\dot{D}_\gamma$, following Equations 3-5. These IM dose rates were also corrected for a theoretical water content of $5 \pm 2\%$ (Table 2) using the equations of Zimmerman (1971). To rapidly generate total $\dot{D}$ estimates, we took the approach of Aitken (1985) whereby water-corrected dose rates are further corrected by multiplication with simplified attenuation factors. This approach is in lieu of the more detailed set of attenuation parameters and calculation steps outlined in Table 2, which are carried out by

software packages like DRAC (Durcan et al., 2015). Aitken (1985) suggests that the water-corrected $\dot{D}_\beta$ of coarse mineral grains that have been etched should be corrected by a factor of 0.9. For the variety of different grain sizes of the theoretical targets in this study, and $\dot{D}_\alpha$, which is a contributor to the total $\dot{D}$ for luminescence dating targets that have not undergone etching, similar mean attenuation factors are provided in Table 3. These mean attenuation factors were calculated using the grain size attenuation data of Brennan et al. (1991). Attenuated $\dot{D}_\alpha$ values were then corrected further for $\alpha$ efficiency using the

$\alpha$-values given in Table 2. Internal $\beta$ dose rates were accounted for in the case of KF or polymineral targets by treating them as constant values for given grain sizes, etch depths and an internal K concentration of $12.5 \pm 0.5\%$ (Huntley and Baril, 1997), as calculated by DRAC using the absorption factors of Guerin et al. (2012) (Table 3). Internal $\alpha$ particle contributions are assumed to be negligible in all cases. The $\dot{D}_c$ was calculated using the equations of Prescott and Hutton (1994) with the same input data as described in Table 2.

**Table 3: Summary of the parameters and assumptions used to calculate rapid dose rates using the IM dose rates predicted based on pXRF K concentrations and training data relationships. Water contents, α-values and $\dot{D}_c$ calculation parameters are the same as Table 2.**

| Input parameter | 180-250 µm quartz | 180-250 µm K-feldspar (etched) | 180-250 µm K-feldspar (not etched) | 4-11 µm polymineral | 4-11 µm quartz |
|---|---|---|---|---|---|
| IM $\dot{D}_\alpha$, $\dot{D}_\beta$ and $\dot{D}_\gamma$ (Gy/ka) | Estimated based on an initial pXRF K measurement using the relationships derived from the training data (Equations 3-5) | | | | |
| $\dot{D}_\alpha$ attenuation[a] | NA | | 0.1±0.01 | 0.9±0.02 | |
| $\dot{D}_\beta$ attenuation[b] | 0.9±0.01 | | | NA[b] | |
| Internal $\dot{D}_\beta$ (Gy/ka)[c] | NA | 0.773±0.138 | | 0.026±0.012 | NA |

[a] Mean attenuation factors were calculated using the data of Brennan et al. (1991).

[b] Mean attenuation factors were calculated using the data of Guerin et al. (2012). A mean $\dot{D}_\beta$ attenuation factor of $0.99 \pm 0.003$

was calculated for the 4-11 µm range, so no correction was applied.

[c] Calculated using DRAC for the grain sizes, etch depths and an internal K concentration given in Table 2.



# 3 Results

## 3.1 Training dataset and relationships

Figure 4 shows the results of IM $\dot{D}_\alpha$, $\dot{D}_\beta$ and $\dot{D}_\gamma$ calculated from the K, U and Th values comprising the 1473 sample training

dataset. Linear regressions fitted between the variables of Equations 3-5 are representative, with $R^2$ values exceeding 0.90 in all cases (Fig. 4), and all models have p-values <0.05, indicating the significance of these relationships at the 95% confidence level. As expected from Equations 3-5, we find positive scaling relationships between K concentration and IM $\dot{D}_\beta$ (Fig. 4a), IM $\dot{D}_\beta$ and IM $\dot{D}_\gamma$ (Fig. 4b) and IM $\dot{D}_\gamma$ and IM $\dot{D}_\alpha$ (Fig. 4c). For ease of interpretation, uncertainties are not shown in Fig. 4 as they are small relative to the dose rate values, with $\dot{D}_\alpha$, $\dot{D}_\beta$ and $\dot{D}_\gamma$ values having mean relative uncertainties of 6.8%, 5.1% and

5.0%, respectively. These uncertainties are a product of the uncertainties of the K, U and Th concentrations used to calculate them, and the uncertainties associated with the radionuclide conversion factors of Guerin et al. (2011). The linear regression equations shown in Fig. 4 form the basis for subsequent rapid dose rate estimation using an initial input of K concentration measured with pXRF.

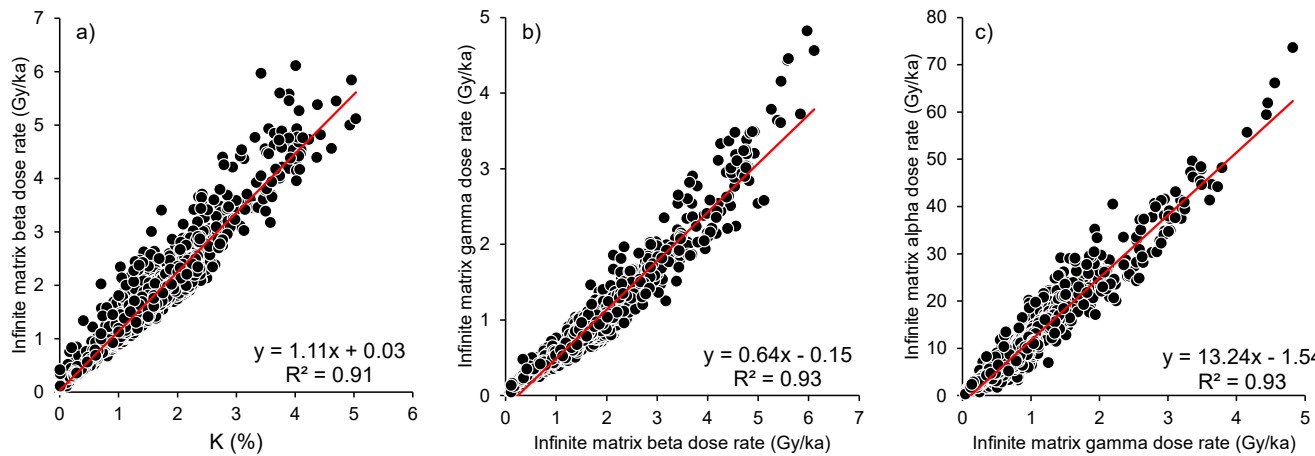

**Figure 4: Training data relationships for: a) K concentration and IM $\dot{D}_\beta$, b) IM $\dot{D}_\beta$ and IM $\dot{D}_\gamma$, and c) IM $\dot{D}_\gamma$ and IM $\dot{D}_\alpha$. Red lines denote the linear trendlines (n=1473). The standard errors for the slopes and intercepts of the regression equations are a) ±0.01 and ±0.02, b) ±0.005 and ±0.01, and c) ±0.1 and ±0.1.**

## 3.2 Portable XRF K concentrations

The results of the training dataset presented in Section 3.1 (Fig. 4) provide linear regression relationships that show that IM

$\dot{D}_\alpha$, $\dot{D}_\beta$ and $\dot{D}_\gamma$ can be estimated from an initial input of the K concentration. Of the 67 samples analysed using pXRF, 66 gave results above the detection limit of the instrument. The only sample that failed to yield a detectable result had a K concentration of 0.02 ± 0.01% (measured with NAA). Based on the training dataset of natural sediment radionuclide contents compiled in this study, sediments with such low K concentrations are rare in nature (Fig. 1; Table 1). Of the 1473 samples included in the training data, only 14 have K concentrations <0.1%, which represents just 1% of the dataset. Portable XRF should, therefore,

be able to provide estimates of K contents in the majority of sedimentary contexts.





Potassium concentrations determined with pXRF show a strong, positive correlation with K concentrations measured using high-precision methods ($R^2 = 0.94$; Fig. 5). The pXRF data are calculated using the mean of three measurements with very small standard deviations relative to mean concentrations (0.0004-0.017%), which demonstrates the consistency of the repeat

measurements. Of the 66 samples that yielded detectable results, 74% have mean pXRF K contents with central values within ±10% of unity with their high-precision counterparts and 91% are within ±20%. The lowest K concentration measured using pXRF was 0.08 ± 0.001%, closely corresponding with a high-precision concentration of 0.13 ± 0.01% measured with ICP-MS. The highest K concentration measured using pXRF was 2.65 ± 0.01%, again closely corresponding with a high-precision concentration of 2.90 ± 0.10% measured using NAA. The slope of the regression equation, 1.09 ± 0.03, shows that the pXRF

instrument generally underestimates the high-precision K concentrations (Fig. 5).

Similarly accurate and reliable results were obtained for the certified reference materials. Blank samples yielded K concentrations consistently below detection limits, indicating that no contamination was present in the pXRF system throughout the measurements.

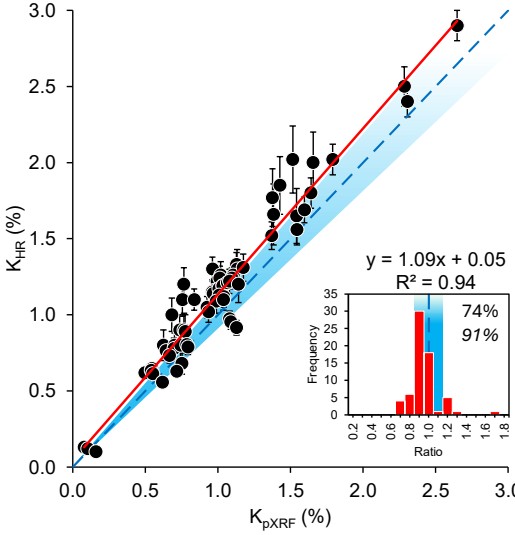


**Figure 5: Potassium (K) concentrations measured using pXRF ($K_{pXRF}$) compared with K concentrations derived from high-precision geochemistry methods ($K_{HP}$). The red line denotes the unweighted linear trendline (n = 66). The standard error of the slope and intercept of the regression equation are ±0.03 and ±0.04, respectively. The dashed blue line and blue shaded area represent unity ± 10%. The inset plot shows frequency distributions of the ratios between $K_{pXRF}$ and $K_{HP}$, with the blue shaded area representing ±**

**10% unity. Percentages show the proportion of central values that fall within this range and values in italics show the proportion of values that fall within ±20% of unity.**





### 3.3 Comparison between rapid and high precision IM and total dose rates

Figure 6 shows the results of calculating IM $\dot{D}_\alpha$, $\dot{D}_\beta$ and $\dot{D}_\gamma$ using rapid pXRF K measurements and the regression relationships derived from the training dataset (Fig. 4), in comparison to calculations based on high-precision radionuclide measurements

and the conversion factors of Guerin et al. (2011). The uncertainties associated with the rapid dose rate values are similar between samples for each emission type. This is because uncertainties are calculated using the margin of error of prediction, the greatest contribution to which are the standard errors associated with the training dataset regressions (Table S2). Due to this, the mean uncertainties associated with each dataset give a representative sense of the magnitude of uncertainty for each IM dose rate (the standard deviation of these mean uncertainties is <0.001 Gy/ka in all cases). The mean uncertainty for rapid

IM $\dot{D}_\alpha$ predictions is the highest (± 3.95 Gy/ka), relative to a mean rapid IM $\dot{D}_\alpha$ of 6.38 ± 4.35 Gy/ka (reported uncertainties around mean values are 1σ). For IM $\dot{D}_\beta$ mean uncertainty is ±0.48 Gy/ka, relative to a mean rapid IM $\dot{D}_\beta$ of 1.17 ± 0.51 Gy/ka. Finally, for IM $\dot{D}_\gamma$ mean uncertainty is ±0.28 Gy/ka, relative to a mean rapid IM $\dot{D}_\gamma$ of 0.60 ± 0.33 Gy/ka.

Significant, positive correlations between rapid and high-precision dose rates are reported for all external dose contributions

(Figure 6a-c). Rapid estimates of IM $\dot{D}_\beta$ based on pXRF K measurements show the closest agreement with their high-precision counterparts, yielding an $R^2$ value of 0.92 (Fig. 6b). Calculating ratios between rapid and high-precision values shows that 74% of central values of rapid IM $\dot{D}_\beta$ results are within ±10% of unity and 91% are within ±20%. When uncertainty ranges are considered, all values fall within ±10% of unity and the regression line follows a very similar trajectory to the 1:1 line (Fig. 6b). The predicted IM $\dot{D}_\gamma$ values, calculated from the predicted IM $\dot{D}_\beta$ results, have a weaker, yet still strong, linear relationship

with high-precision IM $\dot{D}_\gamma$ ($R^2 = 0.57$; Fig. 6c). The slope of the linear trendline for the rapid IM $\dot{D}_\gamma$ vs. high precision IM $\dot{D}_\gamma$ is lower than that of IM $\dot{D}_\beta$, showing that the rapid method generally overestimates IM $\dot{D}_\gamma$ more than IM $\dot{D}_\beta$, and a smaller proportion of central values fall within ±10% (46%) and ±20% (62%) of unity. However, when uncertainty ranges are considered, the majority of predicted IM $\dot{D}_\gamma$ values overlap with the ±10% of unity range. The predicted IM $\dot{D}_\alpha$ values show the weakest relationship with the high precision values ($R^2 = 0.14$; Fig. 6a), with the fewest central values falling within ±10%

(21%) and ±20% (35%) of unity, relative to the other predicted IM external dose rates. The IM $\dot{D}_\alpha$ trendline also has the lowest slope (0.33), showing that, generally, the training relationship overestimates IM $\dot{D}_\alpha$ relative to results calculated using high precision geochemistry (Fig. 6a).

For both the predicted IM $\dot{D}_\alpha$ and IM $\dot{D}_\gamma$, the use of linear regression relationships with negative intercepts (Fig. 4b, 4c) can

result in negative outputs due to low input values (Fig. 6a, c). For the samples analyzed with pXRF in this study, negative IM $\dot{D}_\alpha$ and $\dot{D}_\gamma$ values are predicted for three samples that have K concentrations <0.16%, corresponding with predicted IM $\dot{D}_\beta$ values of <0.21 Gy/ka. Using the regression relationship given in Fig. 4b, negative IM $\dot{D}_\gamma$ results will occur when input IM $\dot{D}_\beta$ is <0.23 Gy/ka, which, using the regression relationship of Fig. 4a, occurs for input K concentrations <0.18%. Using the





regression relationship given in Fig. 4c, negative IM $\dot{D}_\alpha$ will result when input IM $\dot{D}_\gamma$ is <0.12 Gy/ka, which corresponds to an
initial K concentration of <0.35%.

Whilst IM $\dot{D}_\beta$ is generally predicted accurately (within ±10% of unity) by the rapid method, IM $\dot{D}_\alpha$ and IM $\dot{D}_\gamma$ are overestimated
with increasing K contents in sediments, as measured by high-precision methods (Fig. 6d). For IM $\dot{D}_\alpha$, this overestimation is
as much as ∼600% at ∼2.5% K, whereas overestimation is lower for IM $\dot{D}_\gamma$ at only around ∼200% for similar K concentrations
(Fig. 6d). This discrepancy is explained by the fact that the rapid pXRF approach is solely based on K concentration.
Overestimation in IM $\dot{D}_\alpha$ and IM $\dot{D}_\gamma$ are apparent when sediment U and Th contents are low (<1.5 and <5 ppm, respectively),
relative to the mean U and Th contents of sediments in the training dataset (Fig. 6e, f; Table 1), as measured by high precision
methods. The IM $\dot{D}_\gamma$ is contributed by the decay chains of K, U and Th and there is a reasonably strong correlation between K
concentration and IM $\dot{D}_\gamma$ in the training dataset ($R^2 = 0.72$, Fig. S1f). By contrast, IM $\dot{D}_\alpha$ only arises due to U and Th decay,
explaining why IM $\dot{D}_\gamma$ is predicted with greater accuracy than IM $\dot{D}_\alpha$ by the rapid method based solely on K concentration.

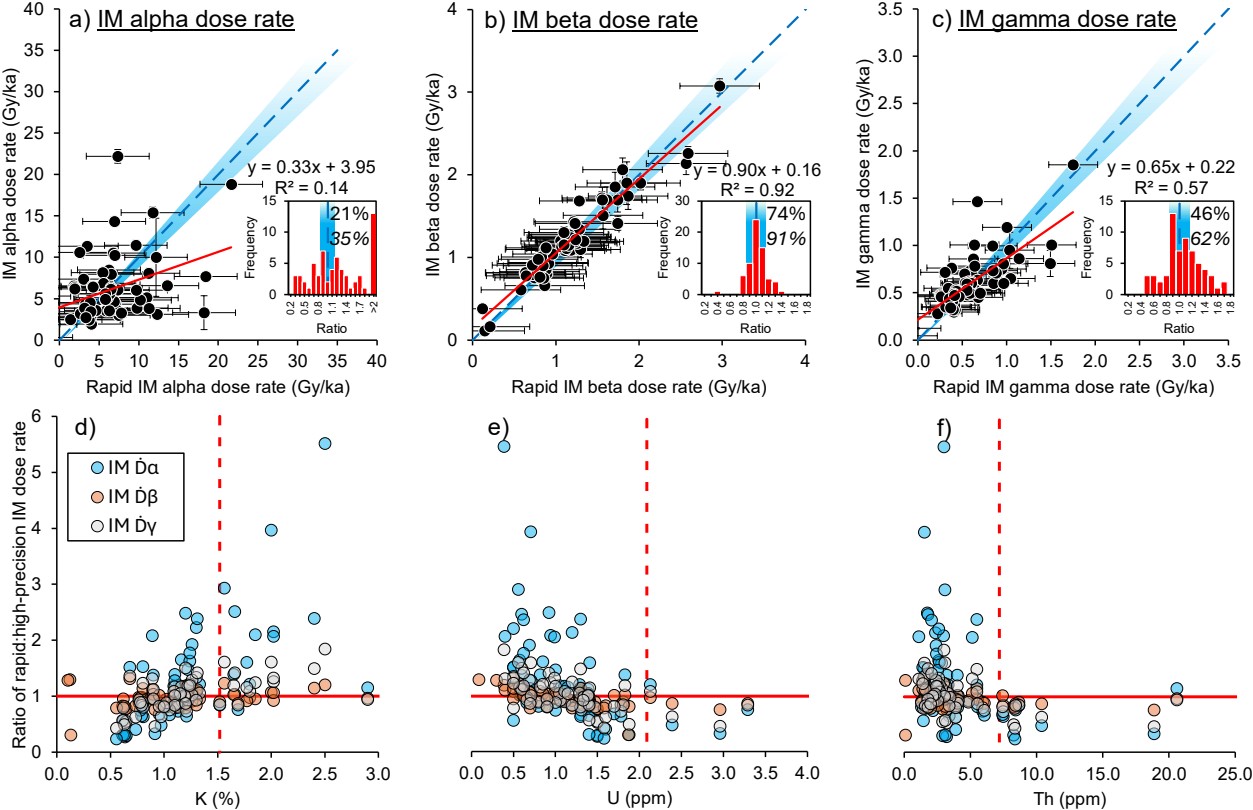

**Figure 6: Results of the IM external dose rates calculated using the training relationships given in Figure 4 based on an initial pXRF measurement of K concentration (x-axes), compared with IM external dose rates calculated from K, U and Th concentrations**



**measured using high-precision geochemistry (y-axes): a) IM $\dot{D}_\alpha$ results, b) IM $\dot{D}_\beta$ results, c) IM $\dot{D}_\gamma$ results. Dashed blue lines and blue shaded areas represent unity ±10%. The red lines denote the linear trendline for each dataset (n = 66 in all cases). The standard errors of the regression slopes and intercepts are a) ±0.10 and ±0.80, b) ±0.03 and ±0.04, and c) ±0.07 and ±0.05. The inset plots show frequency distributions of the ratios between rapid and high precision IM dose rates, with blue shaded areas representing ±10% unity. Percentages show the proportion of central values that fall within this range and values in italics show the proportion of values that fall within ±20% of unity. Panels d), e) and f) show the difference between rapid IM dose rates and high precision dose rates**

**expressed as a ratio plotted against their concentrations of K, U and Th measured with high precision methods, respectively. For ease of interpretation, samples that resulted in negative ratios due to negative dose rates have been omitted. Horizontal red lines show unity and vertical, dashed red lines show the mean concentration of radionuclides in the training dataset (Table 1).**

Figure 7 shows the results of using the rapid pXRF method and simplified attenuation for calculating total $\dot{D}$ for a suite of theoretical dating targets, compared with a standard approach based on high precision geochemistry and more detailed

correction using the DRAC software (Durcan et al., 2015). The rapid approach generally provides good agreement with the high precision approach, with strong positive correlations found in all cases (Fig. 7).

The best agreement between the rapid and high-resolution $\dot{D}$ determinations is found for the coarse-grained targets, which all have $R^2$ values >0.80 and the majority of their rapidly-estimated total $\dot{D}$ values fall within ±10% of unity (Fig. 7a, b, c). Of the

coarse-grained targets, the etched quartz and KF scenarios have the strongest correlations ($R^2 = 0.84$; Fig. 7a, b). These show the best agreement because, due to the assumption that α-irradiated portions of grains have been etched away, the only external dose rates that comprise them are IM $\dot{D}_\beta$ and IM $\dot{D}_\gamma$, which have the strongest correlations with IM dose rates calculated based on high precision geochemistry (Fig.6b, c). For the 180-250 μm quartz example, 56 ± 7% of the total $\dot{D}$ is contributed by the $\dot{D}_\beta$, whilst the $\dot{D}_\gamma$ contributes 32 ± 5% (Table 4). In the 180-250 μm KF (etched) example, the contribution from $\dot{D}_\gamma$ is lower as

a proportion of total $\dot{D}$ (22 ± 5%) due to the contribution of internal β particles (33 ± 1%) (Table 4). Internal dose rate contributions and the $\dot{D}_c$ are the same for both rapidly estimated and high-precision total $\dot{D}$ (Tables 2, 3), meaning that the reduced accuracy in estimating $\dot{D}_\gamma$ using the rapid method is less important in the 180-250 μm KF (etched) scenario, relative to 180-250 μm quartz. This effect explains why the latter scenario yields fewer rapidly estimated total $\dot{D}$ values that are within ±10 and 20% of unity with high-precision total $\dot{D}$ (70% and 88%, respectively; Fig. 7a), relative to the former scenario (85%

and 98%, respectively; Fig. 7b).

By contrast, the larger the contribution of the IM $\dot{D}_\alpha$, the weaker the regression between rapid and high-precision total $\dot{D}$ values. Finer grain-size scenarios (4-11 μm polyminerals and quartz) show more scatter in comparison to high-precision data, due to the incorporation of IM $\dot{D}_\alpha$ components to total $\dot{D}$ on account of them having not been etched (Fig. 7d, e). They have weaker

$R^2$ values (0.66 and 0.77, respectively), although at least 52% of rapidly estimated values still fall within ±10% of unity in both cases and at least 68% are within ±20% of unity (Fig. 7d, e). In the case of the 4-11 μm polymineral scenario, 18 ± 6% of the total $\dot{D}$ is contributed by $\dot{D}_\alpha$, as opposed to only 7 ± 3% in the 4-11 μm quartz scenario (Table 4). Similarly, the incorporation of IM $\dot{D}_\alpha$ into the total $\dot{D}$ of 180-250 μm KF (not etched) example likely results in the correlation with high-precision data being slightly weaker than that of the other coarse-grain scenarios that do not have IM $\dot{D}_\alpha$ contributions (Fig.





7c). However, the IM $\dot{D}_\alpha$ contribution in the 180-250 µm KF (not etched) scenario is, on average, very small (3 ± 1%), so the correlation with high-precision total $\dot{D}$ is stronger than the finer-grained examples ($R^2 = 0.80$; Table 4; Fig. 7c).

In all scenarios, the rapid method typically overestimates total $\dot{D}$, as evidenced by the slopes of the regression equations being <1 (Fig. 7). This is a product of the overestimation that generally results from predicting IM $\dot{D}_\alpha$ and IM $\dot{D}_\gamma$ based solely on K concentration, as discussed above (Fig. 6). Uncertainties are larger for the rapidly estimated total $\dot{D}$ values relative to the high precision data in all scenarios (Fig. 7). The largest source of uncertainty in the rapidly estimated data is the standard error associated with the training regression relationships used to predict IM dose rates based on K concentration. Uncertainties associated with the rapidly predicted IM dose rates are larger than the other sources of uncertainty propagated in quadrature during total $\dot{D}$ calculation arising from water content, attenuation factors, and $\dot{D}_c$ and $\dot{D}_i$ contributions (Tables 2, 3).

Overall, the use of a simplified set of mean attenuation factors in the rapid approach does not result in a significant loss of accuracy with respect to comparing rapid total $\dot{D}$ to high precision $\dot{D}$ for most dating scenarios (Fig. 7). Figure S2 shows total $\dot{D}$, calculated using IM dose rates derived from high-precision K, U and Th measurements but corrected with the simplified attenuation procedure, in comparison to the detailed correction procedure of DRAC. All regressions have an $R^2$ of >0.99 and all values are within ±10% of unity (Fig. S2), demonstrating that the simplified attenuation procedure is contributing little to the discrepancies between rapidly predicted $\dot{D}$ and high-precision $\dot{D}$ (Fig. 7). Inaccuracies in rapidly estimated total $\dot{D}$ are, therefore, more the product of the regression relationships derived from the training dataset (Fig. 4).

**Table 4: Mean percentage contributions and 1σ uncertainties of each constituent external, corrected dose rate to the total environmental dose rate, for the theoretical dating targets shown in Fig. 7 (n=66). These contributions are from the results of high-precision total $\dot{D}$ calculations.**

| Dose contribution (%) | Theoretical luminescence dating target | | | | |
|---|---|---|---|---|---|
| | 180-250 µm quartz | 180-250 µm K-feldspar (etched) | 180-250 µm K-feldspar (not etched) | 4-11 µm polymineral | 4-11 µm quartz |
| $\dot{D}_\alpha$ | 0 | 0 | 3±1 | 18±6 | 7±3 |
| $\dot{D}_\beta$ | 56±7 | 38±8 | 38±7 | 48±7 | 55±7 |
| $\dot{D}_\gamma$ | 32±5 | 22±5 | 21±5 | 24±2 | 28±4 |
| $\dot{D}_i$ | 0 | 33±1 | 32±9 | 1±1 | 0 |
| $\dot{D}_c$ | 12±8 | 7±3 | 7±3 | 9±6 | 10±7 |





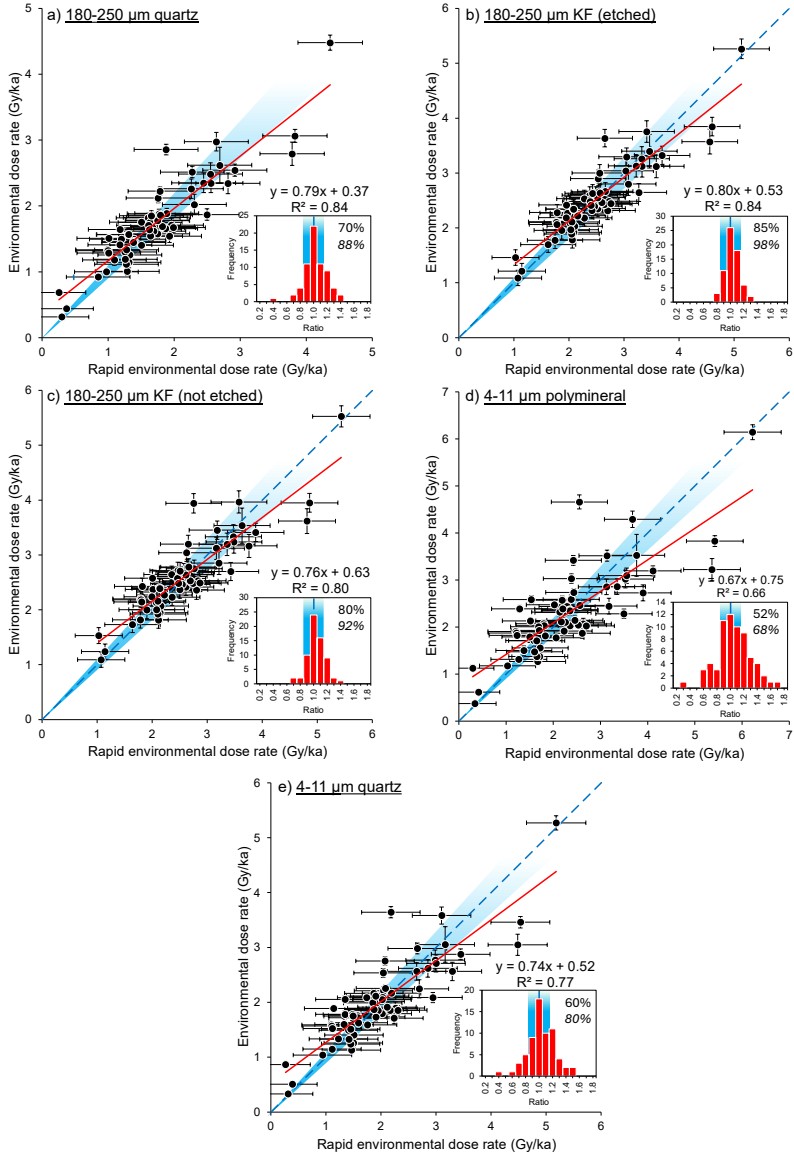

**Figure 7: Total dose rate predicted based on IM dose rates calculated from rapid pXRF measurements of K concentrations and training relationships corrected with simplified attenuation factors (x-axes) and total dose rates calculated using K, U and Th concentrations measured with high-precision geochemistry and full correction in the DRAC software (y-axes) for: a) 180-250 μm quartz, b) 180-250 μm K-feldspar (etched), c) 180-250 μm K-feldspar (not-etched), d) 4-11 μm polyminerals and e) 4-11 μm quartz. The dashed blue lines and blue shaded areas represent unity ±10%, respectively. The red lines denote the linear trendline for each dataset (n=66 in all cases). The standard errors of the regression slopes and intercepts are a) ±0.05 and ±0.08, b) ±0.05 and ±0.12, c) ±0.05 and ±0.13, d) ±0.06 and ±0.16 and e) ±0.05 and ±0.11. The inset plots show frequency distributions of the ratios between rapid and high precision IM dose rates, with blue shaded areas representing ±10% unity. Percentages show the proportion of central values that fall within this range and values in italics show the proportion of values that fall within ±20% of unity.**



## 4 Discussion

### 4.1 Determination of K concentrations using pXRF

Estimates of potassium concentration obtained using pXRF agree very well with high-precision measurements reported for the samples analyzed (Fig. 5), demonstrating both accuracy and reliability, similar to the findings of previous studies using pXRF on sediment samples (e.g., Mejía-Piña et al., 2016; Ou et al., 2022; Zhou et al., 2023), albeit slightly underestimated in the majority of cases. We were able to detect the K concentration of 66 out of 67 samples that had a K concentration >0.02%, with 74% of results falling within ±10% of high-precision geochemistry measurements. This means that pXRF could be used to

accurately determine K contents in most sedimentary contexts, except where K contents are exceptionally low. Given that measurements take only ∼90s per sample, the speed of pXRF analysis enables rapid and large sample throughput in a laboratory setting.

Therefore, it could be possible to make in-situ estimates of K contents for rapid dose rate estimation. However, using a pXRF

system in the field could mean compromising K measurement accuracy in certain scenarios, due to complicating factors like sediment moisture content and heterogeneous grain size, which cause interference (e.g., Nuchdang et al., 2018; Padilla et al., 2019; Rosin et al., 2022). For example, Padilla et al. (2019) show that pXRF underestimates multiple elemental concentrations in a variety of materials with increasing moisture content, relative to expected amounts. Moisture and grain size were controlled in our laboratory experiments by drying and milling sediments prior to analysis, although it is possible that a small pestle and

mortar could be taken into the field to mill sediments in situ. Numerous studies have also developed correction factors to help reduce the influence of moisture on in-situ pXRF measurements, although very site-dependent sediment characteristics mean that the success of these approaches is mixed (e.g., Stockmann et al., 2016; Ribeiro et al., 2018). Trialing pXRF in the field for rapidly estimating dose rates is an important future research avenue.

An important caveat to these findings is that the precision and reliability of elemental measurement can vary between different pXRF instruments (Goodale et al., 2012), so it is important to ensure that instruments are calibrated using reference materials with established elemental concentrations. In this study, all 67 samples analyzed using the pXRF had K contents determined independently using high-precision methods (Fig. 5), although we additionally tested instrument accuracy and contamination using certified reference materials. However, for this approach to be useful in future applications, instrument calibration will

be especially important when K concentrations are not independently known to provide greater confidence in the accuracy and reliability of pXRF measurements.

Other rapid systems for elemental analysis are also available that could be used instead of pXRF for measuring K concentrations in sediments. For instance, XRF core scanners provide rapid, highly spatially resolved K concentrations in

sediment cores from marine and lacustrine environments (e.g., Rothwell and Croudace, 2015), which could be used to derive



dose rates down-core. However, it is important to note that geochemical core scanning is often carried out using intense X-ray beams to provide additional proxies for sediment density and structure, which may destroy natural luminescence signals required for dating (e.g., Davids et al., 2009). Another alternative may be portable laser-induced breakdown spectrometers (pLIBS), which can accurately measure K concentrations with similar rapidity to pXRF (e.g., Lawley et al., 2021). Alternative

approaches to rapidly measure K concentration mean that the approaches developed in this study could be implemented by geoscience and archaeological researchers who may be sampling for trapped charge dating studies in external laboratories, or have access to pOSL units, to help inform sampling strategy or provide range-finder age estimates.

## 4.2 Rapidly estimating environmental dose rates using pXRF

Our results demonstrate that it is possible to estimate a total $\dot{D}$ for range-finder trapped charge dating based on IM $\dot{D}$ derived

from rapidly measured K concentrations alone (Figs 6, 7). We suggest a three-step method for rapidly estimating $\dot{D}$ using pXRF in a laboratory setting (Fig. 8): (1) measure the K concentration of dried, milled sediment using pXRF, taking the mean of triplicate measurements; (2) use the linear equations derived from the training dataset (Fig. 4) to estimate external IM dose rates from pXRF K concentrations; and (3) correct IM dose rates for water content and a simplified set of attenuation factors and add cosmic ray and internal contributions calculated using standard procedures (Table 3). Whilst this approach does not

replace high-precision techniques used for accurate radionuclide and $\dot{D}$ calculation, results show good agreement with $\dot{D}$ based on K, U and Th concentrations measured by high-precision geochemistry and calculated using a more detailed correction procedure (Figs 6, 7). For coarse-grained luminescence dating scenarios, at least 70% of rapid estimates fall within ±10% of unity with their high-precision counterparts, with at least 88% within ±20% of unity (Fig. 7a, b, c).

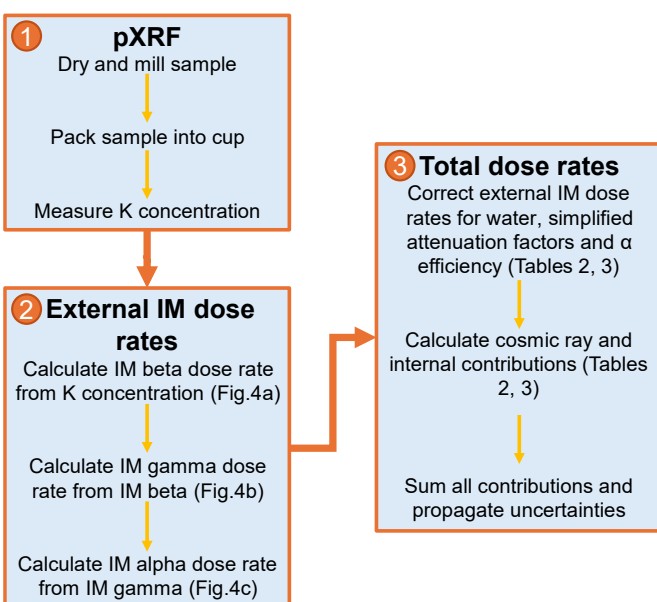

**Figure 8: Flowchart showing the rapid procedure for estimating total environmental dose rates based on pXRF K measurements tested in this study.**



The simple linear regressions used for IM dose rate estimation agree well with previous studies. Ou et al. (2022) derived a linear relationship of IM $\dot{D}_\beta = 1.02K + 0.50$ between the K contents of 61 rock slices and their IM $\dot{D}_\beta$ (measured independently with beta counting). Our relationship of IM $\dot{D}_\beta = 1.11K + 0.03$ derived from 1473 data points is very similar, except with an

intercept much closer to the origin. Previous authors also showed that IM $\dot{D}_\gamma$ can be predicted from IM $\dot{D}_\beta$ using a ratio of ~0.50 (Ankjægaard and Murray, 2007) or 0.59 (Roberts et al., 2009). If the linear relationship determined here for IM $\dot{D}_\beta$ vs. IM $\dot{D}_\gamma$ (Fig. 4b) is forced through the origin, then we derive a very similar ratio from the gradient of the linear trendline, which is 0.58. A second-order polynomial relationship between IM $\dot{D}_\beta$ and IM $\dot{D}_\gamma$ was found to yield a marginally higher $R^2$ value than the linear fit (0.94 vs. 0.93, respectively) using the equation IM $\dot{D}_\gamma = 0.04\text{IM } \dot{D}_\beta{}^2 + 0.44\text{IM } \dot{D}_\beta + 0.03$, as was observed

previously by Ankjægaard and Murray (2007), although these authors also note that their linear ratio of ~0.50 offers nearly equal predictive power. We also found that second order polynomial fits do not increase the strength of correlations for either the K concentration vs. IM $\dot{D}_\beta$ or IM $\dot{D}_\gamma$ vs. IM $\dot{D}_\alpha$ relationships. In lieu of a physical explanation for why these relationships should be fitted with a polynomial, and given the good agreement between our predicted and known total $\dot{D}$ values (Fig. 7), we favor the simple linear regressions for the purpose of rapidly estimating $\dot{D}$.


Out of the predicted IM dose rates, IM $\dot{D}_\beta$ is predicted with the greatest accuracy relative to the high-precision values (Fig. 6b) and IM $\dot{D}_\alpha$ the least (Fig. 6a). This result is unsurprising, given that previous work has shown IM $\dot{D}_\beta$ scales most strongly with K, relative to U and Th (Fig. S1; Ankjægaard and Murray, 2007), whilst IM $\dot{D}_\alpha$ is not physically related to the K decay chain (Guerin et al., 2011). The negative intercepts in the equations relating IM $\dot{D}_\beta$ to IM $\dot{D}_\gamma$ and IM $\dot{D}_\gamma$ to IM $\dot{D}_\alpha$ mean that negative

estimates of IM $\dot{D}_\gamma$ and IM $\dot{D}_\alpha$ can occur at low K concentrations (<0.35% and <0.18%, respectively). Whilst negative dose rates are not physically realistic, only 5% of samples in the training dataset used in this study have K concentrations <0.35% and only 2% are <0.18% (n = 1473). So, negative predictions of IM $\dot{D}_\alpha$ and IM $\dot{D}_\gamma$ are unlikely to occur in most natural sedimentary contexts.

The accuracy of rapidly measured K concentrations using pXRF and the strong relationship derived between K concentrations and IM $\dot{D}_\beta$ could, theoretically, be used to quickly assess $\dot{D}_\beta$ heterogeneity in un-milled sediment and rock samples (e.g., Ou et al., 2022). The significant, positive relationship between IM $\dot{D}_\beta$ and IM $\dot{D}_\gamma$ means that this approach could also be used as a means of rapidly assessing $\dot{D}_\gamma$ heterogeneity (Fig. 4c). However, the weaker correlation found between rapidly estimated and high-precision IM $\dot{D}_\gamma$ (Fig. 6c) means, in practice, that this application would have limited accuracy beyond a rapid, relative

assessment.

Despite the poorer agreement between the predicted and high-precision IM $\dot{D}_\alpha$ values (Fig. 6a), relative to the results for IM $\dot{D}_\beta$ and IM $\dot{D}_\gamma$, the predicted total $\dot{D}$ values for the theoretical dating targets in which IM $\dot{D}_\alpha$ is a factor still yield reasonable $R^2$ values (>0.66) relative to high-precision $\dot{D}$ values (Figs. 7c, d, e). This is because, despite the lower predictive accuracy of this

method for estimating IM $\dot{D}_\alpha$, it typically contributes the least to total high-precision $\dot{D}$ due to the low efficiency with which it



irradiates mineral grains (Table 4). All three IM dose rates are typically overestimated using the relationships derived from the training dataset (Fig. 6). The IM $\dot{D}_\alpha$ is overestimated the most, with overestimation more likely for samples with high K concentrations and/or low U and Th concentrations (Fig. 6d, e, f).

Consequently, total $\dot{D}$ predicted using our rapid method is likely to be more accurate for coarser-grained sediments (e.g., 180-250 μm) that have been etched than for finer-grained sediments (e.g., 4-11 μm) or those that have not been etched, because there is no contribution from α particles in the former scenarios (Porat et al., 2015). This means that our approach is best applied to sedimentary contexts likely to yield coarser size fractions, such as aeolian dune and fluvial deposits (e.g., Wintle, 1993; Wallinga, 2002; Srivastava et al., 2019; Durcan et al., 2019; Wolfe et al., 2023). That said, our results still show
reasonable agreement for finer-grained scenarios and those where etching is not assumed, with at least 52% of rapidly estimated total $\dot{D}$ values falling within ±10% of unity with high-precision values and at least 68% within ±20% (Fig. 7c, d, e). So, this approach still has useful applications to sedimentary contexts that are more likely to be dated using finer grain-size fractions, such as loess and lacustrine deposits (e.g., Singhvi et al., 2001; Roberts, 2008; Fenn et al., 2020; Burrough et al., 2022), or if the laboratory does not routinely etch coarse KF grains (Porat et al., 2015). Our approach could also be adapted to different
grain-size ranges by calculating mean attenuation factors specific to the desired minimum and maximum sizes using attenuation datasets (e.g., Brennan et al., 1991; Guerin et al., 2012).

Lastly, this study only considers the application of this rapid $\dot{D}$ estimation approach to sediment samples as they are most commonly the target of trapped charge dating studies. However, there is growing interest in the application of these
geochronological methods to dating rock surfaces and quantifying their denudation rates (e.g., Sohbati et al., 2015; Jenkins et al., 2018). The pXRF approach to rapid $\dot{D}$ estimation could be usefully applied in solid-rock contexts, especially as internal moisture content is unlikely to be important (although grain-size heterogeneity may be, e.g., Ou et al., 2022). It could offer a non-destructive approach for archaeological and culturally sensitive materials, minimizing the need for invasive sampling (e.g., Gliganic et al., 2021, 2024; Moayed et al., 2023).

**5 Conclusions**

This study provides a proof of concept that a total environmental dose rate, $\dot{D}$, can be estimated using a pXRF measurement of K concentration alone, regression relationships provided by a large training dataset and a simplified set of attenuation factors. This approach is rapid and does not require expensive, specialist facilities. Whilst it is not a replacement for high-precision means of determining $\dot{D}$, it could support trapped charge dating studies by offering a means of rapid range-finder $\dot{D}$
to help inform sampling strategy and generate initial age estimates.





The training dataset utilized is comprised of 1473 sediment samples from around the world with radionuclide concentrations (K, U, Th) measured using high-precision geochemistry. These data represent a large variety of different sedimentary and dosimetry contexts and emphasize the utility of large sample analysis to trapped charge dating studies. The linear regression relationships established based on the training dataset between K concentrations and IM $\dot{D}_\beta$, IM $\dot{D}_\beta$ and IM $\dot{D}_\gamma$, and IM $\dot{D}_\gamma$ and IM $\dot{D}_\alpha$ provide a means of rapidly predicting IM dose rates based on an initial input of K concentration, with strong positive correlations found in all cases.

We found that pXRF provides a rapid and reasonably accurate means of measuring the initial K input to these linear equations, in a controlled laboratory context. We were able to measure K concentrations >0.02% for diverse sediment samples, representing 99% of K concentrations included in the global training dataset. However, questions remain about the accuracy of this method if applied in a field context where grain size and moisture may influence results. The linear relationships used to derive IM dose rate estimates from pXRF K measurements were able to predict IM $\dot{D}_\beta$ with the greatest accuracy with respect to IM $\dot{D}_\beta$ calculated using high-precision K, U and Th data, whilst IM $\dot{D}_\alpha$ was predicted least accurately. Despite inaccuracies in IM $\dot{D}_\alpha$ estimation, good agreement is demonstrated for a range of theoretical luminescence dating targets between total $\dot{D}$ values calculated using rapidly estimated IM dose rates and simplified attenuation procedures, with respect to those calculated using high-precision radionuclide concentrations and more complex attenuation. Agreement between these rapidly predicted dose rates and those calculated with high-precision radionuclides is >70% and >88% within ±10 and 20% of unity, respectively, for coarse-grained quartz and KF scenarios where $\dot{D}_\alpha$ is assumed to be negligible. Even when there are IM $\dot{D}_\alpha$ contributions to the overall dose rate, >52% and 68% of rapidly predicted results fall within ±10 and 20% of unity with high-precision values, respectively. As such, this pXRF-based approach to rapidly estimating dose rates shows promise in a variety of sedimentary settings, even for fine-grained sediments where α particles are likely to contribute more significantly to the $\dot{D}$ of dating targets.

## Data availability

All data are available in the supplementary .csv files.

## Author contribution

SW was responsible for conceptualizing the project, constructing the training dataset, carrying out laboratory measurements, data analysis and the preparation of all figures and the initial draft of this manuscript, under the supervision of MD and OL and with the input of MS and JD. OL and MS provided the sediment samples analyzed in this study. MD, OL, JD and MS provided feedback on the training dataset and laboratory measurements, as well as editing the manuscript.





**Competing interests**

Some authors are members of the editorial board of Geochronology.

**Acknowledgements**

SW would like to thank Shaun Barker and Farhad Bouzari at the University of British Columbia for granting access to the
Mineral Deposit Research Unit's portable X-ray fluorescence spectrometer. The device was purchased using the Natural
Science and Engineering Research Council (NSERC) of Canada's Alliance Grant 570463 "Porphyry fertility and vectoring at
the belt to deposit scale in British Columbia". The Luminescence Dating Laboratory at the University of the Fraser Valley is
supported in part by NSERC Discovery (grant number 311281) and Research Tools and Instrument grants. MD gratefully
acknowledges funding from NSERC's Discovery Grant program, grant number RGPIN-2020-05365.

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
