# Peer review of "Rapid dose rate estimation for trapped charge dating using pXRF measurements of potassium concentration"

_EGUsphere, 2025_

## Author Response (AR1)

**Responses to reviewers' comments**

**Please see author responses to the reviewer comments in bold and italics below.**

**Reviewer 1:**

**We would like to thank the reviewer for their useful comments in improving this manuscript. Please see the original comments from Reviewer 1 below and our responses to them (in bold and italics).**

The manuscript presents a fast dose rate estimation method for trapped charge dating using potassium measurements. The manuscript is well written, but I do have several remarks.

1. First, the interpretation and data analysis are somewhat too optimistic. Correlations might not be that high in a real-world scenario. Second, I was wondering if the term "training" is related to machine learning methods; if not, I would not use it. Third, was the data analyzed using a weighted least squares method, and was the weighting applied to the y or x data axis? The errors on the x-axis are much larger in most of the plots.

*In answer to the three points raised here:*

*1) Yes, in the discussion we highlight some scenarios where our approach may not work so well (e.g., where sediments have not been homogenised prior to pXRF measurement, or in finer-grained settings like lacustrine sediments). We have also made great effort to emphasise the exploratory nature of this technique, rather than it being a replacement for conventional, high-resolution methods for dose rate determination. For instance, the following lines in Section 4.1 where we discuss where K concentrations may not be well estimated by pXRF:*

*"...using a pXRF system in the field could mean compromising K measurement accuracy in certain scenarios, due to complicating factors like sediment moisture content and heterogeneous grain size, which cause interference (e.g., Nuchdang et al., 2018; Padilla et al., 2019; Rosin et al., 2022). For example, Padilla et al. (2019) show that pXRF underestimates multiple elemental concentrations in a variety of materials with increasing moisture content, relative to expected amounts. Moisture and grain size were controlled in our laboratory experiments by drying and milling sediments prior to analysis, although it is possible that a small pestle and mortar could be taken into the field to mill sediments in situ. Numerous studies have also developed correction factors to help reduce the influence of moisture on in-situ pXRF measurements, although very site-dependent sediment characteristics mean that the success of these approaches is mixed (e.g., Stockmann et al., 2016; Ribeiro et al., 2018)."*

*And in Section 4.2 where we discuss specific sedimentary contexts where such an approach may be be less accurate:*

*"The total $\dot{D}$ predicted using our rapid method is likely to be more accurate for coarser-grained sediments (e.g., 180-250 µm) that have been etched than for finer-grained sediments (e.g., 4-11 µm) or those that have not been etched, because there is negligible contribution from α particles in the former scenarios (Porat et al., 2015). This means that our approach is best applied to sedimentary contexts likely to yield coarser size fractions, such as aeolian*

*dune and fluvial deposits (e.g., Wintle, 1993; Wallinga, 2002; Srivastava et al., 2019; Durcan et al., 2019; Wolfe et al., 2023)."*

*2) No, we did not use any machine learning methods. We have removed this terminology to avoid confusion.*

*3) No weighting was applied to the regression analyses (ordinary, least squares regression). The uncertainties are much larger on the x axes (predicted dose rates in the original document) than the y axes (known dose rates in the original document) in our figures due to the uncertainties associated with the equations used to calculate dose rate values.*

> 2. L37 "in Gy per time unit, e.g.," Gy is energy per mass, so "Gy per time" is somewhat misleading. Please remove "Gy per time".

*Thank you, we have removed this. The sentence now reads:*

*"Age calculation using these methods requires two parameters to be quantified: (1) The equivalent dose ($D_e$), the amount of radiation dose absorbed by the mineral throughout the burial period, measured in Gray (Gy); and (2) the environmental dose rate ($\dot{D}$), the rate at which environmental radiation is emitted by the surrounding sediment matrix and received from cosmic rays, measured in Gy/a or Gy/ka."*

> 3. L58-59 "all of which influence radiation emission and absorption." This is not true. For example, radiation emission cannot be changed by grain size. Please remove the fragment "all of which influence radiation emission and absorption."

*Thank you for highlighting this. The sentence now reads:*

*"During dose rate calculation, individual IM dose rates are adjusted for a range of attenuating factors, including grain size, water content, and the effectiveness of α particles to ionize mineral crystals (e.g. Durcan et al., 2015 and references therein)."*

> 4. L87 "De" subscript is missing.

*Thank you for pointing this out, it has been amended.*

> 5. Figure 1 could be replaced by a plot showing dose rates arising from K vs U and Th. This would (or would not) better support the claims.

*These relationships are shown in Fig. S1 and are referred to in the manuscript.*

> 6. L214-228 This interpretation is somewhat too optimistic. From the perspective of this article, we are only measuring K%, so I was wondering if on all plots in Fig. 4, the x-axis should be K% instead.

*Figure 4 is showing a) the relationship between K% and beta dose rate, b) beta dose rate and gamma dose rate and c) gamma dose rate and alpha dose rate: the sequence of relationships used to estimate all three dose contributions from an initial K concentration. There is no alpha emission from K, so we do not think it is useful to plot K vs alpha dose rate (as per Fig. S1), hence the use of an expected gamma dose rate (calculated from a predicted beta dose rate) to estimate alpha. After all, it is not a direct correlation between K% and alpha dose rate that we are using, but rather a set of procedures in between to link the two. It is better to use this stepwise procedure, rather than estimating all dose components from the K input. For example, if one were only to use the relationship between K concentration*

*and the infinite matrix gamma dose rate to predict the gamma dose rate, then this would introduce a lot more scatter in the predictions than if one were to predict the gamma dose rate from the beta dose rate. This effect is shown by the weaker relationship between K and gamma in Figure S1f, relative to the strong relationship between beta and gamma shown in Figure 4b. Of course, these relationships represent an idealised scenario, but they provide us a basis from which to work with a rapid measurement of K as an input.*

7. Figure 5 Would it be possible to add x-error bars? Usually, values that are measured with higher precision should be on the x-axis. This is related to weighting in the least squares method.

*Previously, there were error bars included, however they were just based on the standard deviation of the pXRF measurements, which are very small and thus they were obscured by the points. However, in light of a comment made by Reviewer #2, the uncertainties on the pXRF measurements are now larger due to incorporating the uncertainty associated with the calibration of the measurements based on the certified reference materials. So, the error bars are now visible. We have also swapped all X and Y axes as you suggest so that high-precision measurements are on the X axes and lower precision measurements are on the Y.*

8. Figures 6 and 7 The same comment as above.

*X and Y axes have been swapped.*

9. L103-104 The sentence structure is unclear. Consider rephrasing for better readability.

*Thank you. We have carefully checked through the entire manuscript and made done small rephrasing throughout to improve readability.*

10. L145 Check consistency in terminology when referring to potassium measurement techniques.

*Thank you. We have gone through the full text to make sure our terminology is consistent, where relevant.*

**Reviewer 2:**

**We would like to thank the reviewer for their helpful suggestions and detailed comments in improving this manuscript. Please see the original comments from Reviewer 2 below and our responses to them (in bold and italics).**

Dose rate determination is necessary for most luminescence dating method, and developing tools and method for its determination is essential. This research presents the use of portable XRF, an instrument that is widely available, affordable and precise, for determining K content in sediments, one of the main contributor to dose rate. While this has already been investigated, this study goes further by proposing to estimate the beta dose rate, gamma dose rate and alpha dose rate based on these K measurement and a training relationship obtained thanks to an extended dataset. While the correlation between K content and dose rates seems logical, there is very few attempts to quantify it and even fewer to propose a practical method to do so. Even if some uncertainties on the results are larges (and this is discussed in the paper), this method provide a way to quickly estimate dose rates, with potentially a high spatial resolution and on site. This has the potential to bring very relevant information for targeting samples of interest and a

better understanding of the chrono-stratigraphy of sites, in the same way than portable OSL method already brought (its complementary with the presented method is highlighted in the study).

The manuscript is well constructed, the method and reasoning are well explained. The dataset used for the training relationship is large enough for such a study. Most of the relevant topic are discussed and most of the data are provided. Data about the portable XRF calibration are missing but will hopefully be added.

My main concern is about the use and meaning of some of the statistical tools used for characterizing the goodness of fitting of the training relashionship and of the dose rate obtained by applying it to K measurement with portable XRF. This is developed below with reference to the concerned line and figures. I believe that the methods used are not the right one and I suggest to either switch to a different one that makes more sense, or to justify and discuss of the reasoning behind the application of the current one. It is essential for the study to resolve this point because it has major consequences concerning the uncertainty and precision of the method developed.

Detailed comments:

1. Fig.1 some graph of U/Th and K/U ratios in the same sediments would also provide very useful information to interpret the relationship between K and beta, alpha and gamma dose rate.

**We have added these figures as new panels to Figure 1.**

2. L116: Be careful here: positive correlation does not imply proportionality (which is the meaning of the ∝ symbol). It only mean that the two parameters are moving in the same direction, not that there is a proportionality between them. Please clarify if the reference quoted are demonstrating the proportionality or the only the positive correlation of beta dose rate and K content, and if you are looking for demonstrating their proportionality in the paper.

**Great point. We agree these equations could be misleading and they have been removed. The references cited demonstrate the positive correlation of these variables, not necessarily their proportionality. The text has been edited accordingly:**

**"To estimate $\dot{D}$ based on the K concentration alone, we first establish and test three relationships: 1) $\dot{D}\beta$ is correlated with K concentration, 2) $\dot{D}\gamma$ is correlated with $\dot{D}\beta$, and 3) $\dot{D}\alpha$ is correlated with $\dot{D}\gamma$.**

**Various studies have demonstrated a positive correlation between IM $\dot{D}\beta$ and the K concentration of sediment (Ankjægaard and Murray, 2007; Roberts et al., 2009; Ou et al., 2022). Similarly, Ankjægaard and Murray (2007) showed that IM $\dot{D}\gamma$ can be estimated from IM $\dot{D}\beta$ using either a polynomial regression relationship or a ratio of ~0.50 (determined from the slope of a linear fit), from a large suite of luminescence dating samples and emission-counting methods (n = 3758). Roberts et al. (2009) produced very similar results using linear regression, with a ratio of 0.59 (n = 427). Lastly, IM $\dot{D}\alpha$ should be correlated with IM $\dot{D}\gamma$ because α particles are contributed from the U and Th decay chains (not K), and IM $\dot{D}\gamma$ scales strongly with U and Th concentration (supplementary Fig. S1g, h; Guerin et al., 2011). Therefore, the greater the U and Th concentration, the greater the IM $\dot{D}\gamma$ and, by extension, the IM $\dot{D}\alpha$. Using these principles, we hypothesize that it is possible to estimate IM $\dot{D}\alpha$, $\dot{D}\beta$ and $\dot{D}\gamma$, and therefore $\dot{D}$, from an initial input of the K concentration."**

3. L117 and L121: same issue, positive correlation is different than proportionality. Please clarify which one apply here.

*Please see our response to your previous comment.*

4. L130: This is a nice sampling set. Could you provide some database (spreadsheet, table,...) as supplementary date with their identification, sampling location, radio-element content, type of geology and reference (if published before)? Is there any sample that you rejected for some reason? Are you planning to update your results with additional samples?

*Yes, we have provided a table in the supplementary information with the radionuclide concentrations, location data, infinite matrix dose rates (calculated using the conversion factors of Guerin et al., 2011) and cosmic ray dose rates (calculated using the equations of Prescott and Hutton, 1994). In the updated version of the supplementary information, we have ensured that each sample in this spreadsheet has a reference and the sample code from the original source to aid with traceability. We did not reject any of these data in the analysis. We would like to add to this database for future projects.*

5. L191: Is this water content value from an average of measurement over samples, from a published value, or simply a standard value from experience? Please specify it.

*The water content is purely arbitrary. In the method we propose, the water content is user-defined to correct the rapidly estimated dose rates. So, for the purpose of demonstrating the method, we selected a value of 5±2 just to treat all samples consistently as it does not matter which particular value is selected, as long as the same value is chosen for both the high-precision and rapid dose rate calculation. We have emphasised this point more clearly in the manuscript, as in this part of Section 2.3.2 of the Methods:*

*" These IM dose rates were also corrected for a water content of 5 ± 2% (Table 2) using the equations of Zimmerman (1971). The choice of water content here is purely arbitrary for the purpose of comparison with the high-precision dose rates."*

6. Fig.4: Please be aware that the standard errors of a linear regression represent the uncertainties on the fitting parameters, i.e the range within which these parameters will still represent a good fit of the data. They are not an estimate of well the data point fit to the regression line. Here is a little reasoning to understand what is wrong here: if you were to measure some over datapoints for these graphs, and you managed to measure them with a negligible error, then you would not expect them to align with the regression line within the standard error of its parameters. You would rather expect them to show scatter like the rest of the datapoints, because of the natural variations of radioactive elements (and therefore of dose rates) in the different geological material worldwide.

This mean that if you estimate the uncertainty of your model by using the standard errors of the slope and intercept, you implicitly imply that all measurement should be on the regression line within these errors, and that the data scattering represents the measurement error (in a wide range, including error on sampling, preparation, calibration, measure, etc...). This is difficult to justify considering that the dataset includes samples from very different geological context, and we know that these ratios (beta dose rate / K, beta dose rate / gamma dose rate, gamma dose rate / alpha dose rate) naturally fluctuate from a place to another without being the results of measurement error.

At the opposite (for example), on your fig.5, the standard errors are the right way to estimate the uncertainty, because it is expected that all the data points should align and that the scattering is only due to measurement errors, and not to a natural dispersion.

So, if you want to characterize the variability of the value around the regression, you need to use a metric that indicates how much the data point can scatter around the regression line. The ideal metric for this would be the Root Mean Square Error (RMSE). Please use the RMSE value instead of the standard errors for calculating the uncertainty on the training relationship, or please justify and discuss the use of standard error to characterize the uncertainty on the training relationship.

*Thank you for the detailed explanation, which makes sense. We have exchanged the standard errors for RMSE as you suggest. RMSE is now used to calculate uncertainties in the dose rate calculations. We have made this clear in the updated text of Section 3.1 of the results:*

*" The root mean squared errors (RMSEs) of each relationship were calculated by comparing the predicted variable in each case with the observed variable determined with high-precision chemistry (Fig. S2). For the chosen models, the RMSEs for the predicted IM $\dot{D}_\beta$, IM $\dot{D}_\gamma$, and IM $\dot{D}_\alpha$ values are 0.29, 0.30 and 2.40 Gy/ka, respectively. The regression equations shown in Fig. 4 form the basis for subsequent rapid dose rate estimation using an initial input of K concentration measured with pXRF, with their RMSEs providing uncertainties that are propagated into the final uncertainties on predicted dose rates."*

7. L231: What about the quantification limit rather than the detection limit? If you are using the content of K for estimating the other variables, the quantification limit seems more adapted.

*We have expanded section 2.2 in the methods and section 2 in the supplementary document to explicitly state how the limit of detection (LOD) and limit of quantification (LOQ) are calculated:*

*"The limit of detection (LOD) and limit of quantification (LOQ) for our instrument, with respect to K concentration, were determined as three and ten times the standard deviation of repeat measurements of the CRM with the lowest K concentration respectively (Le Vaillant et al., 2014; Andrew and Barker, 2018; Table S1). The LOD for K in our instrument is 0.015% and the LOQ is 0.049%. Further details of instrument calibration and LOD and LOQ determination are provided in the Supplementary Information."*

*All of the K measurements in this study fall above the LOD and LOQ for our instrument, apart from the same sample as in the previous draft of the manuscript. We have also edited the text in section 3.2 of the Results to discuss the results in terms of both LOD and LOQ:*

*"Of the 67 samples analysed using pXRF, 66 gave results above the LOD of the instrument (LOD = 0.015%). The only sample that failed to yield a detectable result had a K concentration of 0.02 ± 0.01% as measured with NAA. Whilst this low value determined by NAA is in fact higher than the LOD, it also falls beneath the LOQ (LOQ = 0.049%), which may explain why it was not detectable if it was not accurately quantifiable. All of the 66 samples above the LOD were also above the LOQ. Based on the dataset of natural sediment radionuclide contents compiled in this study, sediments with such low K concentrations are rare in nature (Fig. 1; Table 1). Of the 1473 samples included in the dataset, only 14 have K concentrations <0.1%,*

*which represents just 1% of the dataset. Portable XRF should, therefore, be able to provide estimates of K contents in the majority of sedimentary contexts if the LOD and LOQ values as similar to those calculated here."*

8. L235: It is noticeable that even if the K of a sample fell below the detection limit or the quantification limit, that will still allow to infer on the beta dose rate intensity, as well as on the gamma and alpha dose rates (by giving maximum dose rates, that will provide minimum ages) considering the results showed on fig. 4 . So even a non-detection of K in a sample provides useful information.

*Thank you, this is a good point. These details have been added to the discussion in section 4.1 as we think they are better there than in the results:*

*"Even in situations with low K contents, a non-detection could still provide useful information by estimating a maximum dose rate between 0 Gy/ka and the beta dose rate corresponding with the K value determined to be the LOD or LOQ for the specific instrument."*

9. L236 to L245: the 9% of difference between the trend line and the unity line seems typical of a calibration issue. It is unlikely that this could come from the high-precision measurements because you used two different techniques for these measurements. Could you provide more information about the standards used for the pXRF calibration? In particular the K concentration and the type of matrix. For example, if the high K standards have a matrix composition significantly different from the sample you measure, then it can cause a calibration bias. This could help to determine if there is an issue from the calibration, or to exclude this as the source of the observed bias. Please also provide the uncertainty on the pXRF calibration, this is necessary for discussing of the accuracy of the measurements.

*Thank you for highlighting this. This difference between the trend line and the unity line was because, previously, the results had not been properly calibrated due to a misunderstanding on the part of the authors, who are first-time users of pXRF. Through discussion with our colleagues who run the instrument, this has now been rectified using data from the standards that were measured throughout the other samples to check for contamination and the correct method, grounded in the literature. Thank you for your detailed comments, which helped us to spot this! Please see the additional section 2 in the supplementary material and section 2.2 of the methods in the main text detailing the reference standards used and the calibration curve derived (showing the same bias as found in Figure 5 in the previous version of the manuscript) that is now used for correction. Figure 5 has been updated with the properly calibrated measurements, and the trendline is now within uncertainties of unity. Section 2.2 of the Methods now reads:*

*"The results of pXRF analysis were corrected using a linear calibration equation, following previous studies (e.g., Hall et al., 2014; Andrew and Barker, 2018). This calibration equation is the linear relationship between the pXRF-measured K concentrations of the five CRMs and their known K concentrations (Fig. S3)."*

10. Fig.5: Change the Y axis title $K_{HR}$ to $K_{HP}$.

*Changed.*

11. Please provide the uncertainty bars on the pXRF measurement, including the uncertainty on the calibration.

*Previously, this plot only included the standard deviations of the three pXRF measurements per sample, which were very small, hence the points obscured the uncertainty bars. However, we have now calculated uncertainties including the uncertainty from the calibration. This has increased the size of the error bars, which are plotted and visible on the revised Figure 5.*

12. On the frequency graph, it is unclear what the 74% and 91% values are referring too. Please either add a comment in the figure description or simply delete them (as I don't think they are necessary to understand the frequency graph, and they are already given in the main text).

*We have deleted the frequency graph from this figure as we agree they were unnecessary.*

13. I am surprised to see that some of the uncertainty on the high-precision K measurements are quite high (up to 10%). Could you provide some information about that? Please also indicate if the uncertainties are provided at 1 or 2 sigma.

*Great question, we are not sure as to the answer, which would require delving into the original sources. The uncertainties on the high-precision K measurements used in the calculations are as reported in the sources or from the commercial laboratories who carried out the analyses. Further details about the high-precision radionuclide contents, including uncertainties, references and sample locations are available in the supplementary information as a .csv file.*

14. L262: As explained for fig.4, in this case the standard errors of the regression are not a good estimate of the uncertainty of the observed relationships in the training data set. This is because the data scattering is not due to measurement errors but to the natural variability of sediments. You should use RMSE as uncertainty.

*Please see our response to your earlier comment (6). The uncertainties are now calculated using RMSE.*

15. Could you also specify if the uncertainty on the pXRF measurement of table S2 includes the pXRF calibration uncertainty? If not, please add this uncertainty to your calculations.

*Thank you. The pXRF measurement uncertainties now incorporate the calibration uncertainty. These uncertainties have been updated in table S2.*

16. L264: "the standard deviation of these mean uncertainties is <0.001 Gy/ka in all cases". It is unclear what you mean by this, and what does it mean for your results. Could you elaborate this further?

*Upon reading it back, this section was needlessly confusing and does not add much, so has been deleted. We wanted to make the point that the uncertainties are very similar between predicted dose rates, but this is evident from looking at the figures. We now have two, simpler sentences:*

*"The uncertainties associated with the rapid dose rate values are similar between samples for each emission type. This is because uncertainties incorporate the RMSEs of the*

***predictive models (Fig. 4), which are the same for each sample, as well as smaller uncertainties contributed by the input K concentrations measured using the pXRF (Fig. 5)."***

17. L269: Could you actually present some quantitative value of these correlation, for example by calculating their Pearson correlation coefficient (PCC)? That will allow the reader to evaluate how significant are the correlations. In the case of the alpha dose rate, I am not so sure that this can be called a good correlation (but the PCC may tell overwise).

***We have calculated the PCC, as you suggest, to quantify the strength of the correlations. They are presented alongside trendline equations and R2 values in Figures 6 and 7 and used in text when the strength of the association between the x and y variables is being discussed. For example, as in Section 3.3 of the Results:***

***"Rapid estimates of IM $\dot{D}_\beta$ based on pXRF K measurements show the strongest positive correlation with their high-precision counterparts (r = 0.96) and the closest agreement relative to the unity line ($R^2$ = 0.75; Fig. 6b)."***

18. L269 to L282: Here there is a wrong use of statistics, or a misunderstanding of their meaning: if you want to quantify the agreement between the rapid estimate and the high precision estimate of the dose rates, then you should to calculate the $R^2$ of your datapoints relative to the unity line, not the one relative to the datapoint trendline. Here is an absurd reasoning: imagine all your datapoints are perfectly aligned (negligible scattering) around a trendline with a slope of 0.33 (like you have for the alpha dose trendline). Then your $R^2$ relative to the trendline would be 1 (perfect match), however there would be a very poor agreement between rapid estimates of dose rate and high precision estimates.

***Thank you for this explanation. The R2 values presented in figures 6 and 7 are now calculated relative to the unity line and are consequently weaker than previously reported. The text has been adjusted throughout the results accordingly.***

19. L280 to L282: considering the high scattering beyond uncertainties and low $R^2$, the most likely conclusion would rather be that a linear trend is simply not representative of your datapoints. But this may change once you recalculate the uncertainties on your fast measurement and training relationship according to the previous comments.

    Besides, if you look at the red trendline of fig. 6a compared to the unity line, you can see that the supposed relationship actually underestimates the dose rate when it is below 5 Gy/ka.

***This text has been adjusted according to the new results, as per the response to your above comment.***

20. L284 to L290: Could you actually calculate the standard errors on parameters of the regression lines, and check if there are compatible with the unity line (i.e that the slope is within 2 standard error value of the value 1 and that the intercept is within 2 standard error value of 0). If it is, that would simply mean that the differences between trendline and unity line can be explained by the scattering of the datapoints.

***Using the polynomial fit for the beta vs. gamma relationship has resolved the issue of generating negative gamma and alpha doses, at least in terms of the range of values considered here, as suggested by Reviewer 3. Nonetheless, the standard errors of the***

*parameters of the trendlines are given in the caption of Figure 6 and the text in this section has been updated accordingly.*

    21. L291 to L300: This makes sense, and that could allow you to calculate the reliability of the method depending on the measured K content (using the data from the training set for example). This would be a very useful information for the method you developed.

*Thank you. Reviewer #3 also made some similar comments about testing the method with the training data as it represents a broader range of values. We have added new panels to Figure 4 (panels d-f) that show the residuals of using the training dataset as inputs to these equations (both linear and polynomial), which covers a larger range of K concentrations and dose rates than the samples included in this study. We have discussed these results in greater detail in Section 3.1 of the Results.*

    22. L316 to L320: same than for L269 to L282, you cannot test the agreement by calculating the $R^2$ between the datapoints and their trendline. For that, you need to calculate the $R^2$ between the datapoints and the unity line. However, you can test if the trendline parameters are compatible with the unity line within their standard error (see comment for L284 to line 290).

*The text and figures have been adjusted with R2 values calculated relative to the unity line, as well as Pearson's correlation coefficients calculated relative to the trendlines, as per your earlier comment (17).*

    23. L335: same comment than for L316 to L320 and for L269 to L282

*Please see the response to your previous comment.*

    24. L343 to L349 and fig.7: It is not always an overestimation because for lower dose rate you have an underestimation (the red trendline is above the unity line).

*Good point. Note that the correctly calibrated pXRF data and swapping the axes around, relative to the previous version of the manuscript, means that the two versions are not compatible here. Previously, there was an underestimation at low doses, now there is an overestimation. However, we have made the text here clearer to specify that, with these updated data, this overestimation occurs below around 5 Gy/ka:*

*"In all scenarios, the rapid method typically overestimates total $\dot{D}$ for instances where the high-precision calculated dose rate is < 5 Gy/ka, as evidenced by the slopes of the regression equations being <1 (Fig. 7). This is a product of the overestimation that generally results from overestimations of IM $\dot{D}_\beta$, as well as overestimations of low IM $\dot{D}_\alpha$ and IM $\dot{D}_\gamma$ values, as discussed above (Fig. 6). The convergence of the trendline with the unity line at 5 Gy/ka in each scenario suggests that higher K concentrations would result in overestimations being more likely, although beyond the range of the 66 samples measured here."*

    25. It would be useful to test if the trendline parameters are compatible with the unity line within their standard error (see comment for L284 to line 290). That will help to see if their difference can be explained by the uncertainty on the linear regression.

*The newly-calculated standard errors on the trendline parameters are given in the caption of Figure 7 and we have updated the text in this section to read (following on from the text in the previous response):*

*"...However, in all cases the slopes of the trendlines shown in Figure 7 are within two standard errors (given in the caption of Fig. 6) of the unity line. The intercepts are more dispersed, with the coarse-grained scenarios all having intercepts either within two standard errors of the unity line or very close (within 0.01 Gy/ka of two standard errors), whilst the fine-grained scenarios are not within or close to two standard errors of unity."*

26. You seem to forget an important source of error here: the sampling bias. Every dating laboratory have regional areas of predilection, depending on the focus of their studies and the projects they have (for example, some laboratories have a high focus on South Africa, while other would have high focus on North America). This could be a factor significant enough for creating a bias between the dataset available at your lab and the more worldwide dataset used to build the training relationship. I invite you to generate training relationship for the different parts of the world for the which you have data, to see if significant difference in trends can be observed. Of course, this represent some work that may be beyond the focus of your current manuscript, but you should at least discuss of this potential sampling bias.

*This is a very good point and we have acknowledged it in section 4.2 of the discussion as we felt it was more suited there:*

*"These discrepancies between fitting parameters reported in different studies may likely be explained by different sample sizes or different sampling biases, namely the geological origin of samples. In this study, the majority of the 67 samples that we tested using this rapid approach were sourced from western North America, the radionuclide contents of which will be dependent on their specific source geology. Therefore, the results we demonstrate may not be representative of samples from other parts of the world, given differences in the geological origins of sediment. Whilst beyond the scope of this study, it will be important to test the approach proposed here on samples from other locations to determine the influence of local factors on prediction uncertainties. Similarly, testing the potential sensitivity of the models used to rapidly predict dose rates (Fig. 4) to specific regions and their different ratios of radionuclide concentrations is also an important next step."*

27. L354: same comment about the calculation of $R^2$ than for L269 to L282, although this will not change significantly your result here.

*Thanks. Same responses as those above regarding R2. You are correct though, the results are unchanged.*

28. L337: do not forget the measurement error part in the uncertainty.

*Thank you. We assume this refers to L357 in the original manuscript, but we have incorporated measured uncertainty in these calculations and noted this in the text:*

*"Uncertainties are larger for the rapidly estimated total $\dot{D}$ values relative to the high precision data in all scenarios (Fig. 7). The largest sources of uncertainty in the rapidly estimated data are the RMSEs associated with the regression relationships used to predict IM dose rates and the measurement uncertainties on the pXRF K concentration and its calibration."*

29. L375: I would say that "very well" is a little overstated and should be removed. From fig. 5 there seems to be a bias that resemble to a calibration issue and should be tested or discussed further.

***The measurements now do agree very well by applying the calibration properly! See Section 2.2 of the methods and the supplementary information for more details on the calibration.***

30. L384 to L394: This point occurred in my mind while reading about your lab setting. It is indeed very important and I am please to see it discussed. You could add that the in-lab testing that you did is a necessary step before undergoing any field study on the subject.

***Good point. We have added this to the text in Section 4.1:***

***"Whilst our laboratory experiment serves as a necessary first step, trialling pXRF in the field for estimating dose rates in a range of different conditions is an important future research goal."***

31. L399: please provide (in supplementary data) data about your calibration (type of standard certification, their matrix, their K content) and the uncertainty associated with the calibration.

***Thank you for this comment. Information about the calibration and materials used are now provided in section 2.2 of the Methods and in the section 2 of the supplementary information.***

32. L427 to L433: You should add that the differences between the observed fitting parameter could be likely explained by the size of the different datasets and the sampling bias related to the nature or origin of the samples in each dataset.

***We have added this in combination with the previous comment that you made regarding sampling bias with respect to the different origins of samples (26). Please see our response above.***

33. L444 to L449: It is likely that, once you based your uncertainty on the RMSE instead of the standard error of the parameters like recommended, these negative intercepts will be compatible with the 0 value within uncertainty. I believe they are simply an artifact of the natural dispersion of radioactive elements and dose rate in geological samples around the world.

***This is less of an issue now that we are using the polynomial fit to estimate gamma dose rates, as suggested by Reviewer 3. However, this is a good point and we have added it as a comment at the end of the paragraph:***

***"The negative intercept we observe may be the result of the natural dispersion of radionuclides in different sedimentary contexts, as well as uncertainties in their conversion to dose rates. Given that the RMSE of the IM $\dot{D}_\gamma$ vs. IM $\dot{D}_\alpha$ relationship is 2.40 Gy/ka (Fig.4c), negative estimates would likely be within uncertainties of 0 Gy/ka for most K concentrations."***

34. L451: You should also quote Nathan R. Jankowski, Zenobia Jacobs (2018): Beta dose variability and its spatial contextualisation in samples used for optical dating: An empirical approach to examining beta microsimetry, Quaternary Geochronology 44, https://doi.org/10.1016/j.quageo.2017.08.005. They used pXRF for K spatially

resolved K measurement in geological sample, with even an attempt to quantify U and Th with this instrument.

***This is a very useful addition, thank you. We have also cited it in the introduction to exemplify the point that U is especially hard to detect with pXRF.***

35. L467: negligible contribution will be more correct than "no contribution"

***Changed.***

36. L478 to L485: You seem to forget that in the case of exposure surface dating, there is no need for dose rate calculation (the dose rate is not a part of the dating equation. It will only be useful in the case of burial surface dating, when the bleached surface is buried again and start re-accumulating charges under the effect of radiation.

***Thanks for spotting this; we mixed our terms up. The references used here were indeed examples of rock burial dating, rather than exposure. We have changed the terminology used.***

**Reviewer 3:**

**We would like to thank the reviewer for their helpful suggestions and detailed comments in improving this manuscript. Please see the original comments from the reviewer below and our responses to them (in bold and italics).**

The paper addresses an important part of luminescence dating: rapid determination of dose rates, potentially on-site screening use. The paper is well written and I think it contributes an important tool to the luminescence dating community, however I have a few remarks.

1. I am guessing that the "training" referred to in the paper is machine learning, but this is never specified. The pXRF is operated in two beam mode, but no explanation is given for what this means, nor is any data on the instrument or its settings provided.

***Reviewer 1 raised the same comment about the use of the word 'training'. No machine learning was used, just the simple regression models as laid out in the paper. 'Training' has been removed throughout to avoid this confusion.***

***Details on the pXRF instrument have been greatly expanded in Section 2.2 of the methods and in the supplementary material. Please see our responses to your more detailed comments about the pXRF analyses below.***

2. One part which I feel is missing, is how the model works in the extremes. Based on the training set I assume that there are samples with low K but higher concentrations of Th and U. It would support the overall application of basing everything on measurements K content to show how the predictions fit in such cases. For now I feel that there is strong reliance on $R^2$ values to determine how the method works with most of the datapoints clustered in a very narrow range. This is obviously not something the authors can control, but given the extensive dataset used for the training, some investigation over a broader dose rate range would help strengthen the paper.

***Thank you for the comment, which echoes suggestions made by Reviewer #2. An additional set of panels have been added to Fig. 4 (d-f), showing the residuals of these predictions***

*across this large dataset to better understand how the model performs in the extremes, which is discussed in an updated Section 3.1 of the Results.*

3. Equations 3, 4, 5: ∝ does not directly imply a positive correlation, rather it is used to show proportionality between two or more variables. I'd suggest adjusting the text accordingly.

*Thank you. This comment was also made by reviewer 2 and the text has been adjusted in section 2.1 of the methods:*

*"Various studies have demonstrated a positive correlation between IM $\dot{D}\beta$ and the K concentration of sediment (Ankjægaard and Murray, 2007; Roberts et al., 2009; Ou et al., 2022). Similarly, Ankjægaard and Murray (2007) showed that IM $\dot{D}\gamma$ can be estimated from IM $\dot{D}\beta$ using either a polynomial regression relationship or a ratio of ~0.50 (determined from the slope of a linear fit), from a large suite of luminescence dating samples and emission-counting methods (n = 3758). Roberts et al. (2009) produced very similar results using linear regression, with a ratio of 0.59 (n = 427). Lastly, IM $\dot{D}\alpha$ should be correlated with IM $\dot{D}\gamma$ because α particles are contributed from the U and Th decay chains (not K), and IM $\dot{D}\gamma$ scales strongly with U and Th concentration (supplementary Fig. S1g, h; Guerin et al., 2011). Therefore, the greater the U and Th concentration, the greater the IM $\dot{D}\gamma$ and, by extension, the IM $\dot{D}\alpha$. Using these principles, we hypothesize that it is possible to estimate IM $\dot{D}\alpha$, $\dot{D}\beta$ and $\dot{D}\gamma$, and therefore $\dot{D}$, from an initial input of the K concentration."*

4. Figure 4: Did you compare the K beta dose rates when using the conversion factors of Creswell et al. (2018) as well, since this should be more accurate? It should increase the K beta dose rate about 7%, which is rather substantial. It would also affect later figures.

*Thank you for this comment. Initially, no, we did not try the conversion factors of Creswell et al. (2018) as they are not yet available in the DRAC software used for dose rate calculation. However, we have now tested the method with the Creswell et al. conversion factors. We found that the regression relationships are changed only minutely by changing the conversion factors used to calculate IM dose rates from the large radionuclide dataset. Furthermore, if one uses the regression relationships derived from Cresswell et al.'s conversion factors, rather than those of Guerin et al., to calculate IM dose rates, there agree very closely with those based on the regression relationships using Guerin et al.'s conversion factors. We have provided new figures in the supplementary material showing 1) the regression relationships that result from data calculated using Cresswell et al.'s conversion factors (Fig.S2), and 2) the IM dose rates that result from the use of these regressions in comparison to those based on the Guerin et al. (2011) conversion factors (Fig.S3). These figures provide a parallel to Figures 4 and 6 in the main text, calculated using the conversion factors of Guerin et al. (2011). So, users can use either set of equations as they choose. We have also noted this in the manuscript, directing readers to view these supplementary figures.*

5. Figure 5: Any ideas why the pXRF underestimates K? Was it the same for reference materials?

*Thanks for spotting this! Previously, these data had not been calibrated correctly due to a misunderstanding in the use of the instrument (we are new users). This has now been rectified through discussions with the experts who run the instrument, and we have a much better agreement between known and pXRF K values. Please see our responses to the comments made about this by Reviewer #2 (comments 7 and 9), the expanded section 2.2*

*of the manuscript and additional section 2 in the supplementary detailing the calibration of measurements. For example, Section 2.2 of the Methods now reads:*

*"The results of pXRF analysis were corrected using a linear calibration equation, following previous studies (e.g., Hall et al., 2014; Andrew and Barker, 2018). This calibration equation is the linear relationship between the pXRF-measured K concentrations of the five CRMs and their known K concentrations (Fig. S3)."*

6. Figure 6: It would be nice to see comparison of higher IM gamma dose rates as the data shown is still in the region where Fig. 4b appears to be described by a linear function. It would also show how suitable the choice of linear vs second order is for the training.

   L432: Here I disagree that the linear fit should be used in lieu of a physical explanation as Figure 4b clearly shows that the relationship between IM beta dose rate and IM gamma dose rate is not linear, especially at higher IM beta dose rates. Assuming that you always have the same proportions of K, Th, and U and only the absolute activities are changing, then it would be linear, but that seems like an overly ideal situation. Since most of the datapoints are appear to be clustered between 0-3 Gy/ka, the $R^2$ likely wouldn't change much if you use a second order polynomial, but at higher IM beta dose rates I'm guessing your fit would be a lot better.

*These two comments are answered together as they have similar themes. Thanks for prompting us to look again at the fit; you make a good point and we agree. We have switched to the use of a polynomial fit to describe the beta vs. gamma relationship, as you suggest, for the following reasons:*

*•  It has a higher R2 value than the linear fit, suggesting that it does provide a better representation                                                          of                                    the                                    data.*

*•  It does not have a negative intercept, reducing the issues of estimating negative gamma, and     subsequent     alpha     dose     rates     that     are     not     physically     realistic.*

*•  A nonlinear trend has been observed previously in the literature using a larger dataset (n = 3758) generated by comparing beta counts to gamma spectrometry measurements (Ankjægaard and Murray, 2007).*

*The new panels added to Figure 4 (d-f) show the residuals of the predicted vs expected dose rates expressed as a percentage of the expected, for both linear and polynomial models. The polynomial residuals are, overall, smaller than the linear ones for the gamma dose rate, but very similar for beta and alpha, as visual inspection of the trends in Fig. 4a-c suggest. Interestingly, the best improvements are for samples with low IM gamma dose rates. At the low end, the linear model sometimes results in negative gamma and alpha contributions due to its negative intercept: something that is also physically impossible. However, at the high end, the linear equation actually appears to perform very slightly better than the polynomial (black points are closer to the red line than the white points at high values in Fig 4e) for these data. We present both linear and polynomial equations for estimating each dose rate so that users can decide which they prefer (e.g., they may want to use linear for gamma if they expect high values).*

**References:**

Ankjærgaard, C. and Murray, A.S., 2007. Total beta and gamma dose rates in trapped charge dating based on beta counting. Radiation Measurements, 42(3), pp.352-359.

---

## Author Response (AR2)

**Responses to editor's comments**

**Please see author responses to the editor's comments in bold and italics below.**

**We would like to thank the Associate Editor, Sumiko Tsukamoto, for their useful comments in improving this manuscript. Please see the original comments from the Associate Editor below and our responses to them (in bold and italics).**

Thank you very much for submitting your responses and the revised version of the paper. The manuscript has been improved significantly.

I have only one remaining concern regarding Fig. 4. There are several remarks about the intercept of the fitted line. However, these seem unnecessary if the fitting is constrained through the origin (for both linear and polynomial fittings), which is also the approach adopted in previous studies. Reviewer 2 also noted that the intercept does not carry a clear physical meaning. In fact, for the data of Fig. 4b you calculated the slope by forcing the fit through the origin for comparison. Could you consider applying this approach consistently, or alternatively provide a clear justification for retaining the current method of fitting?

***Thank you for this comment. We have decided to leave the fits shown in Figure 4 as they are, retaining the intercepts. As in the previous work cited in the discussion (Ankjægaard and Murray, 2007), when the fits are forced through the origin, we observe a decrease in the predictive accuracy of the models for the predicted alpha and gamma dose rates, whilst the beta dose rates are the same. Therefore, as we are aiming for the greatest predictive power based on our large dataset, this justifies retaining the intercepts. The same argument was made for the choice of a second order polynomial fit for predicting the gamma dose rate from the beta dose rate, prompted by Martin Autzen's comments, so this rationale is consistent.***

***I have added a comment to the section of the discussion where the comparison between fitting choices is made (Line 564):***

***'If we use a linear fit forced through the origin for these data then the ratio of IM $\dot{D}_\gamma$ to IM $\dot{D}_\beta$ would be 0.58, which agrees very closely with previous findings of 0.50 (Ankjægaard and Murray, 2007) and 0.59 (Roberts et al., 2009). However, we find that there is a poorer agreement with unity for the relationships between the data calculated without the intercepts and high precision dose rates for both estimated IM $\dot{D}_\gamma$ ($R^2$ = 0.51) and IM $\dot{D}_\alpha$ ($R^2$ = -0.29), relative to the estimates calculated using the intercepts shown in Figure 6, whilst the accuracy of IM $\dot{D}_\beta$ estimates are the same. Ankjægaard and Murray (2007) also found that using a model fitted through the origin also resulted in a slight reduction of predictive power when estimating IM $\dot{D}_\gamma$. Whilst both sets of results are within uncertainties, we suggest that the intercepts be retained.'***

Other minor suggestions are listed below.

-Line 16: I would remove "accurately".

***Changed, thank you.***

-Line 24-25: Could you comment about gamma?

***A comment has been added (Line 25):***

*'The regression equations can predict external beta dose rates to a good degree of accuracy based on K content alone, whilst external gamma dose rates are predicted less accurately and external alpha dose rates are predicted the least accurately.'*

- Guerin should be Guérin (throughout the text)

*Changed, thank you.*

-Rizza et al. (2024) (line 82) is not listed in the reference list.

*Added, thank you.*

- Consider combining Tables 2 and 3. I see why they are separate, but it could be easier for readers to see all these parameters in one table.

*The tables have been combined into just Table 2, thank you.*